# Decomposing stimulus-specific sensory neural information via diffusion models

**Steeve Laquitaine**[*]
Institut de la Vision, INSERM, CNRS
Sorbonne Université
17 Rue Moreau, Paris 75012

**Simone Azeglio**[*]
Institut de la Vision, INSERM, CNRS, & Laboratoire des Systèmes Perceptifs
Sorbonne University & École Normale Supérieure - PSL
17 Rue Moreau, Paris 75012 & 29 Rue d'Ulm Paris 75005

**Carlo Paris**
Institut de la Vision, INSERM, CNRS
Sorbonne Université
17 Rue Moreau, Paris 75012

**Ulisse Ferrari**[†]
Institut de la Vision, INSERM, CNRS
Sorbonne Université
17 Rue Moreau, Paris 75012

**Matthew Chalk**[†]
Institut de la Vision, INSERM, CNRS
Sorbonne Université
17 Rue Moreau, Paris 75012
`matthew.chalk@inserm.fr`

## Abstract

To understand sensory coding, we must ask not only how much information neurons encode, but also what that information is about. This requires decomposing mutual information into contributions from individual stimuli and stimulus features—a fundamentally ill-posed problem with infinitely many possible solutions. We address this by introducing three core axioms—*additivity*, *positivity*, and *locality*—that any meaningful stimulus-wise decomposition should satisfy. We then derive a decomposition that meets all three criteria and remains tractable for high-dimensional stimuli. Our decomposition can be efficiently estimated using diffusion models, allowing for scaling up to complex, structured and naturalistic stimuli. Applied to a model of visual neurons, our method quantifies how specific stimuli and features contribute to encoded information. Our approach provides a scalable, interpretable tool for probing representations in both biological and artificial neural systems.

## 1 Introduction

A central question in sensory neuroscience is *how much*, but also *what* information neurons transmit about the world. While Shannon's information theory provides a principled framework to quantify the amount of information neurons encode about *all* stimuli, it does not reveal *which* stimuli contribute most, or *what* stimulus features are encoded [25, 2, 8]. As a concrete example, it is known that

---

[*]Equal contribution.
[†]Equal senior author contribution.

39th Conference on Neural Information Processing Systems (NeurIPS 2025).

neurons in the early visual cortex are 'sensitive' to stimuli in a small region of space (their receptive field). However, it is not clear how such simple intuitions carry to more complex scenarios, e.g. with large, noisy & non-linear population of neurons and high-dimensional stimuli.

Several previous measures of neural sensitivity have been proposed. For example, the Fisher information quantifies the sensitivity of neural responses to infinitesimal stimulus perturbations [23, 3, 31, 18, 9]. However, as the Fisher is not a valid decomposition of the mutual information it cannot say how different stimuli contribute to the total encoded information. On the other hand, previous works have proposed stimulus dependent decompositions of mutual information, which define a function $I(x)$ such that $I(R; X) = \mathbb{E}[I(x)]$ [8, 4, 5, 17]. However, this decomposition is inherently ill-posed: infinitely many functions $I(x)$ satisfy the constraint, with no principled way to select among them. Further, different decompositions behave in qualitatively different ways, making it hard to interpret what are they are telling us. Finally, most proposed decompositions are computationally intractable for the high-dimensional stimuli and non-linear encoding models relevant for neuroscience.

To resolve these limitations, we propose a set of axioms that any stimulus specific and feature-specific information decomposition should satisfy in order to serve as a meaningful and interpretable measure of neural sensitivity. These axioms formalize intuitive desiderata: that the information assigned to each stimulus, and stimulus feature, should be non-negative, and additive with respect to repeated measurements. We also require the decomposition to respect a form of *locality*: changes in how a neuron responds to a stimulus $x$ should not affect the information attributed to a distant stimulus $x'$. Finally, the attribution must be insensitive to irrelevant features, which do not contribute to the total information. Together, these constraints ensure that the decomposition is both interpretable and theoretically grounded.

We show that existing decompositions violate one or more of these axioms, limiting their interpretability and use as information theoretic measures of neural sensitivity. We then introduce a novel decomposition that satisfies all of our axioms. It generalizes Fisher information by capturing neural sensitivity to both infinitesimal and finite stimulus perturbations. Moreover, it supports further decomposition across individual stimulus features (e.g., image pixels), enabling fine-grained analysis of neural representations.

Beyond satisfying our theoretical axioms, our decomposition is computationally tractable for large neural populations and high-dimensional naturalistic stimuli, through the use of diffusion models. We demonstrate the power of our method by quantifying the information encoded by a model of visual neurons about individual images and pixels. Our approach uncovers aspects of the neural code that are not picked up by standard methods, such as the Fisher information, and opens the door to similar analyses in higher-order sensory areas, and artificial neural networks.

## 2 Desired properties of stimulus-specific decomposition of information

We aim to decompose the mutual information $I(R; X)$ between a stimulus $X$ and a neural response $R$ into local attributions $I(x)$, assigning to each stimulus $x$ a measure of its contribution to the total information. By construction, such a decomposition must satisfy:

- **Axiom 1: Completeness.** The average of local attributions must recover the total mutual information:
$$I(R; X) = \mathbb{E}_X[I(x)]. \tag{1}$$

This constraint alone does not uniquely determine the function $I(x)$, as many decompositions satisfy completeness. To further constrain the attribution, we impose additional desiderata that reflect desirable properties of local information measures.

First, for $I(x)$ to serve as an *interpretable*, stimulus-specific measure of neural sensitivity, it should satisfy a *locality principle*: perturbations to the likelihood &/or prior in a neighbourhood of $x_0$ should have a vanishing influence on $I(x)$ for distant stimuli $x \neq x_0$. Without this property, changes in $I(x)$ may reflect changes in neural sensitivity to any stimuli, undermining interpretability.

- **Axiom 2: Locality.** Let $p(R, X)$ and $\tilde{p}(R, X)$ be two joint distributions over the response, $R$, and stimulus, $X$. Suppose there exists finite $\epsilon > 0$ and $x_0$ such that:
$$p(R, x) = \tilde{p}(R, x) \quad \text{for all } x \notin B_\epsilon(x_0), \tag{2}$$

where $B_\epsilon(x_0)$ is the open ball of radius $\epsilon$ centered at $x_0$. Assuming $X$ has infinite support on an unbounded or semi-infinite domain, we require that, for every $\delta > 0$ there should exist some finite value $d$, such that:

$$\left| I(x) - \tilde{I}(x) \right| \leq \delta \quad \text{for all } x \notin B_d(x_0), \tag{3}$$

where $I(x)$ and $\tilde{I}(x)$ denote the corresponding information decompositions derived from $p(R, x)$ and $\tilde{p}(R, X)$, respectively. That is, local perturbations to the likelihood or prior near $x_0$ should not affect the information assigned to distant stimuli, $x$.

Mutual information is globally non-negative: $I(R; X) \geq 0$. To preserve this property in our decomposition, we require the pointwise contributions $I(x)$ to be non-negative as well. Intuitively, observing a neural response can only refine our beliefs about the stimulus—it cannot undo information. In addition, negative attributions can harm interpretability, since they can cancel out, obscuring how different stimuli contribute to the total information.

- **Axiom 3: Positivity.** Local information attributions must be non-negative:

$$I(x) \geq 0 \quad \text{for all} \quad x \in X. \tag{4}$$

Finally, Shannon 1948 posited additivity as a fundamental property of mutual information: the total information from multiple sources should equal the sum of their individual contributions. We extend this principle pointwise, requiring that information combine additively across measurements.

- **Axiom 4: Additivity.** For two responses $R$ and $R'$, the local attribution should decompose as:
$$I_{R,R'}(x) = I_R(x) + I_{R'|R}(x), \quad \text{for all} \quad x \in X \tag{5}$$
where we have renamed $I(x)$ as $I_R(x)$ here, to make explicit its dependence on the response, $R$. $I_{R'|R}(x)$ is the conditional pointwise information from $R'$ given $R$, and $I_{R,R'}(x)$ denotes the pointwise information from observing both responses. By construction, we require: $I(R'; X|R) = E_x \left[ I_{R'|R}(x) \right]$ and $I(R, R'; X) = E_x \left[ I_{R,R'}(x) \right]$.

*Remark 1: Local data processing inequality.* Any decomposition that fulfils both additivity and positivity, as stated above, also obeys a local form of the data processing inequality. That is, post-processing should not increase information, even at the level of the individual attributions. Formally,

$$\text{If} \quad X \to R \to R', \quad \text{then} \quad I_R(x) \geq I_{R'}(x) \quad \text{for all} \quad x \in X \tag{6}$$

To prove this we use additivity to write $I_{R',R}(x)$ in two ways:

$$I_R(x) + I_{R'|R}(x) = I_{R'}(x) + I_{R|R'}(x) \tag{7}$$

Positivity gives $I_{R|R'}(x) \geq 0$ for all $x$, so:

$$I_R(x) + I_{R'|R}(x) \geq I_{R'}(x) \tag{8}$$

Now, if $R'$ is independent of $X$ given $R$, then $I(R'; X|R) = \mathbb{E}_X[I_{R'|R}(x)] = 0$. Since if $I_{R'|R}(x) \geq 0$ pointwise, this implies $I_{R'|R}(x) = 0$ for all $x$. Substituting into the inequality above yields the desired result: $I_R(x) \geq I_{R'}(x)$.

*Remark 2: Invariance to invertible transformations.* The data processing inequality implies that local information attributions are invariant under invertible transformations of the response variable. That is, for any invertible function $\phi(R)$, we have:

$$I_{\phi(R)}(x) = I_R(x). \tag{9}$$

This follows from the fact that $I(R; X) = I(\phi(R); X)$, and hence $\mathbb{E}_X[I_{\phi(R)}(x)] = \mathbb{E}_X[I_R(x)]$. Supposing, for contradiction, that $I_{\phi(R)}(x) < I_R(x)$ for some $x$, equality of expectations would require $I_{\phi(R)}(x) > I_R(x)$ for some other $x$, violating the pointwise data processing inequality. This highlights why our axioms are important for interpretability: they imply that $I(x)$ quantifies how information is transmitted through the system $X \to R$, independently of how the responses are parameterized.

Table 1: Satisfaction of axioms by different information-theoretic measures of neural sensitivity. Only our proposed decomposition, $I_{local}$, fulfils all the axioms.

| Axiom | $\mathcal{J}(x)$ | $I_{sp}(x)$ | $I_{SSI}(x)$ | $I_{surp}(x)$ | $I_{CiSSI}(x)$ | $I_{local}(x)$ |
|---|---|---|---|---|---|---|
| Completeness | ✗ | ✓ | ✓ | ✓ | ✓ | ✓ |
| Locality | ✓ | ✗ | ✗ | ✗ | ✗ | ✓ |
| Positivity | ✓ | ✗ | ✗ | ✓ | ✓ | ✓ |
| Additivity | ✓ | ✓ | ✓ | ✓ | ✓ | ✓ |

Table 2: Satisfaction of axioms by different information-theoretic measures of neural sensitivity. Only our proposed decomposition, $I_{local}$, fulfils all the axioms.

| Axiom | $J_{Fisher}(x)$ | $I_{sp}(x)$ | $I_{SSI}(x)$ | $I_{surp}(x)$ | $I_{CiSSI}(x)$ |
|---|---|---|---|---|---|
| Completeness | ✗ | ✓ | ✓ | ✓ | ✓ |
| Locality | ✓ | ✗ | ✗ | ✗ | ✗ |
| Positivity | ✓ | ✗ | ✗ | ✓ | ✓ |
| Additivity | ✓ | ✓ | ✓ | ✓ | ✓ |

## 3 Previous stimulus-dependent decompositions

Several previous works have proposed decompositions of mutual information into stimulus-specific contributions, such as the stimulus-specific information $I_{SSI}$ [4], the specific information $I_{sp}$, the stimulus-specific surprise $I_{surp}$, [8], and the coordinate-invariant stimulus-specific information $I_{CiSSI}$ [17]:

$$I_{sp}(x) = H(R) - H(R|x) \tag{10}$$

$$I_{SSI}(x) = H(X) - \mathbb{E}_{R|x}[H(X|R = r)] \tag{11}$$

$$I_{surp}(x) = D_{\mathrm{KL}}(p(R|x)\|p(R)) \tag{12}$$

$$I_{CiSSI}(x) = \mathbb{E}_{R|x}[D_{\mathrm{KL}}(p(X|R = r)\|p(X))]. \tag{13}$$

While these decompositions satisfy some of our proposed axioms (Table 2), none satisfy locality. This is because they all depend on global terms such as the marginal response distribution $p(r) = \int p(r|x')p(x')\,dx'$, or the posterior $p(x|r) = p(r|x)p(x)/\int p(r|x')p(x')\,dx'$, both of which can be influenced by changes to the likelihood &/or prior for any stimulus. Further, the requirement that $I_{CiSSI}(x)$ is invariant to invertible transformations of $x$ is incompatible with locality; both axioms can't be fulfilled simultaneously. As discussed above, this limits the interpretability of previous decompositions as measures of neural sensitivity, since a non-zero attribution at $x$ could be due to changes in neural sensitivity anywhere in the stimulus space.

Unlike the above decompositions, the Fisher information is inherently local. However, while previous authors found a relation between the Fisher and mutual information [3, 29], this only holds approximately, and in certain limits (e.g. low-noise). Therefore, the Fisher information cannot be used to quantify how different stimuli contribute to the total encoded information (i.e. it fails the **completeness** axiom).

Recently Kong et al. 2024 used diffusion models to decompose the mutual information into contributions from both the stimulus, $x$, *and* the response, $r$. Two different decompositions, $I(r, x)$, were proposed. However, averaging these over $p(R|x)$ does not give stimulus-wise decompositions that fulfil our axioms (Appendix B). In one case, we obtain $I_{surp}(x)$ (Eqn 12), which is non-local; in the other case, the decomposition is not additive, which is a fundamental information theoretic constraint.

In the following we propose a new stimulus-wise decomposition of the mutual information which fulfils all our axioms, combining the advantages of the Fisher information (locality, positivity & additivity) while being a valid decomposition of the mutual information.

## 4 Diffusion-based information decomposition

We first derive an expression for the mutual information, $I(R; X)$, that can be used to construct a stimulus-dependent decomposition that fulfils all of the above axioms.

### 4.1 Exact relation between Shannon information and Fisher information

We consider a population of neurons which show a response, $R$, to a stimulus, $X$, with probability $p(R|X)$. We then consider a noise-corrupted version of the stimulus, $X_\gamma = X + \sqrt{\gamma}Z$, where $Z \sim \mathcal{N}(0, I)$. From the fundamental theorem of calculus we can write:

$$-\int_{\gamma_0}^{\infty} \frac{dI(X_\gamma; R)}{d\gamma} d\gamma = I(R; X_{\gamma_0}) - \lim_{\gamma \to \infty} I(R; X_\gamma) = I(R; X_{\gamma_0}), \tag{14}$$

since in the limit $\gamma \to \infty$, $X_\gamma$ is just noise, and thus $I(R; X_\gamma) \to 0$. It follows that,

$$I(R; X_{\gamma_0}) = -\int_{\gamma_0}^{\infty} \frac{dI(X_\gamma; R)}{d\gamma} d\gamma = \int_{\gamma_0}^{\infty} \left( \mathbb{E}_{p(R)} \left[ \frac{dh(X_\gamma|r)}{d\gamma} \right] - \frac{dh(X_\gamma)}{d\gamma} \right) d\gamma \tag{15}$$

where $h(X_\gamma) \equiv -\mathbb{E}_{p(X_\gamma)}[\log p(X_\gamma)]$ and $h(X_\gamma|r) \equiv -\mathbb{E}_{p(X_\gamma|r)}[\log p(X_\gamma|r)]$. Assuming that $p(X)$ and $p(X|R)$ have finite second order moments, we can apply de Bruijn's identity for all $\gamma > 0$, to obtain,

$$
\begin{aligned}
I(R; X_{\gamma_0}) &= -\frac{1}{2}\text{Trace} \int_{\gamma_0}^{\infty} \mathbb{E}_{p(X_\gamma, R)} \left[ \nabla_{x_\gamma}^2 \log p(X_\gamma|R) - \nabla_{x_\gamma}^2 \log p(X_\gamma) \right] d\gamma \\
&= -\frac{1}{2}\text{Trace} \int_{\gamma_0}^{\infty} \mathbb{E}_{p(X_\gamma, R)} \left[ \nabla_{x_\gamma}^2 \log p(R|X_\gamma) \right] d\gamma \\
&= \frac{1}{2}\text{Trace} \int_{\gamma_0}^{\infty} \mathbb{E}_{p(X_\gamma)} \left[ \mathcal{J}(X_\gamma) \right] d\gamma
\end{aligned}
\tag{16}
$$

where $\mathcal{J}(x_\gamma)$ is the Fisher information with respect to a noise-corrupted stimulus, $X_\gamma$. Finally, since $\lim_{\gamma_0 \to 0} I(R; X_{\gamma_0}) = I(R; X)$, and $\text{Trace}(E_{p(X_\gamma)}[J(X_\gamma)])$ is always non-negative, monotone convergence yields:

$$I(R; X) = \frac{1}{2}\text{Trace} \int_0^{\infty} \mathbb{E}_{p(X_\gamma)} \left[ \mathcal{J}(X_\gamma) \right] d\gamma. \tag{17}$$

This is a general result that holds for both discrete and continuous $X$ and $R$, so long as $p(X)$ and $p(X|R)$ have finite first and second moments.

The above identity provides a direct relation between the mutual information $I(R; X)$ and the Fisher information, $\mathcal{J}(x_\gamma)$. Further, it also admits a natural interpretation in terms of de-noising diffusion models trained to predict $X$ from a noisy observation $X_\gamma$. To see this, first we use an alternative formulation of the Fisher information, in terms of the mean-squared score:

$$
\begin{aligned}
I(R; X) &= \frac{1}{2} \int_0^{\infty} \mathbb{E}_{p(X_\gamma, R)} \left[ \left\| \nabla_{x_\gamma} \log p(R|X_\gamma) \right\|^2 \right] d\gamma \\
&= \frac{1}{2} \int_0^{\infty} \mathbb{E}_{p(X_\gamma, R)} \left[ \left\| \nabla_{x_\gamma} \log p(X_\gamma|R) - \nabla_{x_\gamma} \log p(X_\gamma) \right\|^2 \right] d\gamma
\end{aligned}
\tag{18}
$$

Next, from Tweedie's formula [20, 14] we have:

$$I(R; X) = \int_0^{\infty} \frac{1}{2\gamma^2} \mathbb{E}_{p(X_\gamma, R)} \left[ \left\| \hat{x}(X_\gamma, R) - \hat{x}(X_\gamma) \right\|^2 \right] d\gamma, \tag{19}$$

where $\hat{x}(x_\gamma) = \mathbb{E}[X|x_\gamma]$ and $\hat{x}(x_\gamma, r) = \mathbb{E}[X|x_\gamma, r]$. These conditional means can be approximated using denoising diffusion models trained to sample from the prior, $p(X)$, and posterior $p(X \mid R)$, respectively [26].

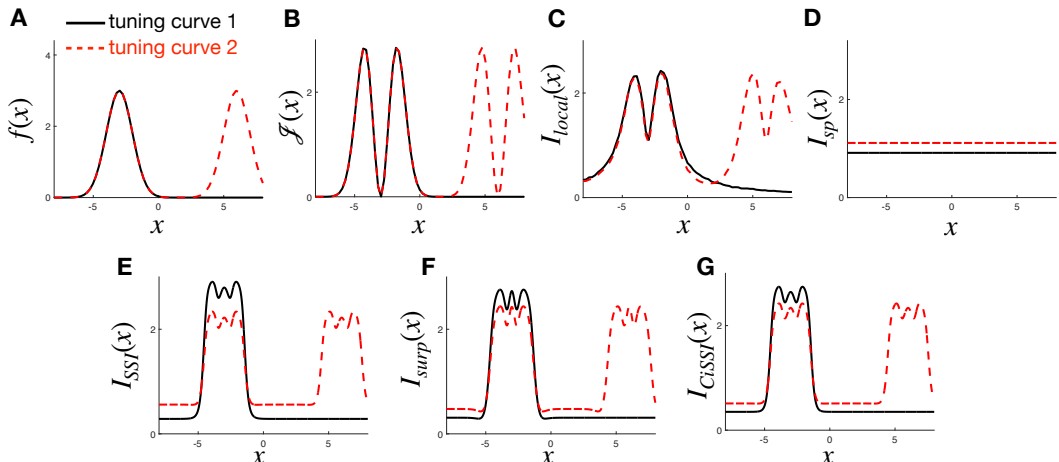

Figure 1: **Demonstration of locality.** (**A**) A neuron responds to a 1D stimulus $x$ with tuning curve $f(x)$. We compare two cases: (i) tuning changes only near $x = -3$ (black), and (ii) additional changes near $x = 6$ (red dashed). (**B–C**) The Fisher information, $\mathcal{J}(x)$ and local information $I_{\text{local}}(x)$ converge as we go far from the region where the two tuning curves differ. (**C–F**) For other decompositions (Eqs. 10–13), local changes in the tuning curve result in non-local changes to the information attribution, for all $x$.

## 4.2 Local stimulus-specific information

Building on the integral representation of mutual information in Eqn 17, we define a stimulus-specific decomposition:

$$I_{local}(x) = \frac{1}{2} \sum_i \int_0^\infty \mathbb{E}_{p(X_\gamma|x)} \left[ \mathcal{J}_{ii}(X_\gamma) \right] \, d\gamma, \tag{20}$$

where $X_\gamma = x + \sqrt{\gamma}Z$, with $Z \sim \mathcal{N}(0, I)$, and $\mathcal{J}_{ii}(x_\gamma) = \mathbb{E}_{p(R|x_\gamma)} \left[ -\nabla^2_{(x_\gamma)_i} \log p(R \mid x_\gamma) \right]$ is the $i^{th}$ diagonal element of the the Fisher information matrix, evaluated at $x_\gamma$. This expression parallels Eq. 17, with the key distinction that the expectation is now conditioned on a fixed stimulus $x$, rather than averaging over the full stimulus distribution [3]. Consequently, the decomposition satisfies the **completeness axiom**, since by construction, $I(R; X) = \mathbb{E}\left[I_{local}(x)\right]$.

Next we outline why our decomposition fulfils the **locality axiom** (for the formal proof, see Appendix A). Recall that the Fisher information matrix $\mathcal{J}(x)$ characterizes the local curvature of the log-likelihood, $\log p(R \mid x)$, and thus quantifies the local sensitivity of the response to changes in the stimulus [23]. The term $\mathbb{E}_{X_\gamma|x}\left[\mathcal{J}(X_\gamma)\right]$ generalizes this notion, measuring neural sensitivity when we only observe noise-perturbed versions of the stimulus, $X_\gamma \sim \mathcal{N}(x, \gamma I)$. For finite $\gamma$, this term is dominated by values of $X_\gamma$ close to $x$, and thus, it depends only on the local shape of the likelihood and prior around $x$. It receives a vanishingly small contribution from changes to the likelihood and/or prior for distant stimuli $x'$, if $\|x' - x\| \gg \sqrt{\gamma}$. Meanwhile, as $\gamma \to \infty$, $X_\gamma$ becomes pure noise and thus $E_{p(X_\gamma|x)}\left[\mathcal{J}(X_\gamma)\right] \to 0$ for all $x$. Taken together, this implies that our decomposition, obtained by integrating $E_{p(X_\gamma|x)}\left[\mathcal{J}(X_\gamma)\right]$ over all $\gamma > 0$, satisfies the **locality axiom**: local perturbations to $p(R, x)$ affect $I_{local}(x')$ only for nearby $x'$, while their influence vanishes as $\|x' - x\| \to \infty$.

The remaining axioms follow directly from standard properties of the Fisher information matrix. **Positivity** follows from the fact that the Fisher information is positive semi-definite. **Additivity** follows from the identity $\mathcal{J}_{R',R}(x_\gamma) = \mathcal{J}_R(x_\gamma) + \mathcal{J}_{R'|R}(x_\gamma)$ [32]. Both properties are preserved when we average the Fisher information over $p(X_\gamma|x)$ and integrate over $\gamma > 0$, to obtain $I_{local}(x)$.

---

[3]Note that the exchange of integrals over $\gamma$ and $x_\gamma$ to go from Eqn 17 to Eqn 20 is justified by the fact that the integrand, $\mathcal{J}_{ii}(x_\gamma)$, is Lebesgue integrable, since it is always positive and integrates to a finite value.

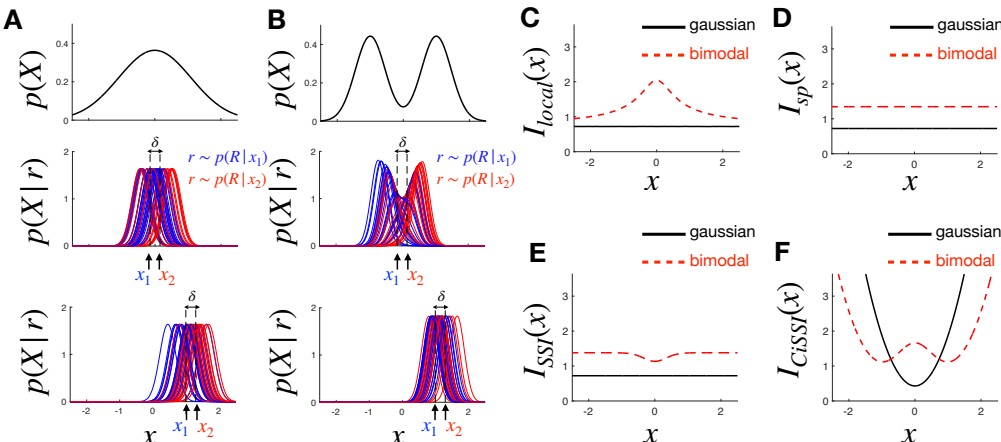

Figure 2: **Effect of prior.** (A) A gaussian prior (top) with a neuron responding as $r = x + \text{noise}$. Middle and bottom: posterior $p(X \mid r)$ given $r \sim p(R \mid x_1)$ (blue) or $r \sim p(R \mid x_2)$ (red), for stimuli $x_1, x_2$ separated by $\delta$. The posterior is equally sensitive to $x$ near 0 (middle) than near 1 (bottom). (B) Same as A, but with bimodal prior (top). The posterior is more sensitive to $x$ near 0 (middle) than near 1 (bottom). (C) The local information peaks near $x = 0$ for the bimodal prior (red dashed), where the posterior is most sensitive, and is flat for a Gaussian prior (black), where posterior shape is constant. (D–F) Other decompositions behave differently: (D) $I_{\text{sp}}$ is flat for both priors; (E) $I_{\text{SSI}}$ is minimal near $x = 0$ for the bimodal prior; (F) $I_{\text{CiSSI}}$ is quadratic under a Gaussian prior. $I_{surp}$ (not shown) behaves similarly to $I_{\text{CiSSI}}$.

## 4.3 Feature-Wise Decomposition

We assume the stimulus $x$ is a vector of image features $x_i$ (e.g. image pixels). Given that the local stimulus information (Eqn. 20) is expressed as a sum over diagonal elements of the Fisher information matrix, it is natural to decompose $I_{local}(x)$ into feature-wise contributions $I_i(x)$.

As with the stimulus-wise decomposition, this problem is ill-posed: infinitely many decompositions exist in theory. However, the same axioms constrain the feature-wise decomposition. To satisfy **additivity**, $I_i(x)$ must be a linear combination of Fisher diagonal terms: $I_i(x) = \frac{1}{2} \sum_j a_{ij} \int_0^\infty \mathbb{E}_{X_\gamma|x}[\mathcal{J}_{jj}(X_\gamma)] \, d\gamma$. **Completeness** requires $\sum_i a_{ij} = 1$, while **positivity** enforces $a_{ij} \geq 0$. Finally, to fully specify the weights, $a_{ij}$, we need to introduce one further axiom, which ensures that the attribution $I_i(x)$ is zero for irrelevant stimulus features, $X_i$, which are independent of the response, $R$.

- **Axiom 5: Insensitivity to irrelevant features.** If $X_i$ is independent of $R$, then $I_i(x) = 0$ $\forall x \in X$.

For this axiom to hold, we need to set $a_{ij} = \delta_{ij}$, so that $I_i(x) = \frac{1}{2} \int_0^\infty \mathbb{E}_{X_\gamma|x}[\mathcal{J}_{ii}(X_\gamma)] \, d\gamma$. If, on the contrary, $a_{ij} \neq \delta_{ij}$, then a neuron's sensitivity to other features (i.e. $\mathcal{J}_{jj}(x) > 0$) could 'leak over' to make $I_i(x) > 0$ even when $X_i$ is independent of $R$, violating the axiom.

## 5 Results

### 5.1 Locality

To illustrate the implication of the locality axiom, we analyzed the responses of a model neuron to a one-dimensional stimulus drawn from a Gaussian prior. The neuron's response was modeled as a Gaussian random variable with mean $f(x)$ and fixed standard deviation. We compared two tuning curves: one with a single peak at $x = -3$ (Fig. 1A, black), and another with an additional peak at $x = 6$ (Fig. 1A, red).

With gaussian noise, Fisher information scales with $f'(x)^2$, and thus peaked where the tuning curves were steepest (Fig. 1B). Similar qualitative behaviour was observed for $I_{\text{local}}(x)$ (Fig 1C). Crucially,

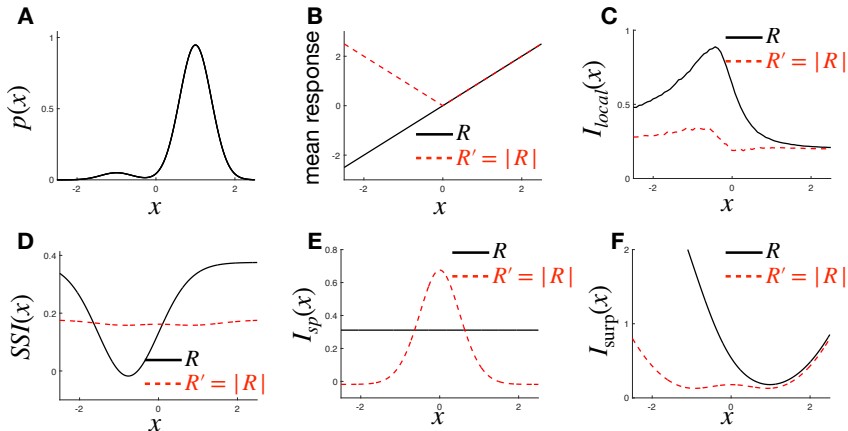

Figure 3: **Local data processing inequality.** (**A**) Bimodal prior. (**B**) Neural responses: $r = x +$ noise; transformed response $r' = |r|$. The non-invertible transform reduces information. (**C**) Our decomposition satisfies the pointwise data processing inequality (DPI): $I_{R'}(x)$ (red) is always less than or equal to $I_R(x)$ (black). (**D–F**) $I_{surp}$ also satisfies pointwise DPI, while $I_{SSI}$ and $I_{sp}$ do not. $I_{CiSSI}$ (not shown) behaves similarly to $I_{surp}$.

the difference in $I_{local}(x)$ between the two tuning curves vanished outside the region where they differ, consistent with the locality axiom. In contrast, existing attribution methods (Fig. 1D–G) showed global sensitivity: adding a second peak at $x = 6$ altered the attributed information across the entire stimulus space, including far from the added feature.

## 5.2 Effect of Prior

We next examined how $I_{local}(x)$ responds to changes in the shape of the stimulus prior. For this, we considered two different stimulus priors: a zero-mean gaussian, and a bimodal mixture of two gaussians with peaks at $x = -1$ and $x = 1$ (Fig. 2A-B, upper panels). To isolate the effect of the prior, we used a simple linear-Gaussian likelihood model: $r \sim \mathcal{N}(x, \sigma^2)$. Under this model, the neuron's sensitivity is uniform across all stimuli, so any variation in attributed information must arise solely from the prior.

Intuitively, one can assess neural sensitivity by measuring how much the posterior distribution $p(X \mid r)$ changes, on average, in response to small perturbations in the stimulus $x$. With the bimodal prior (Fig. 2B), the posterior is highly sensitive near $x = 0$, where the two modes compete (Fig. 2B, middle panel), and relatively stable near the modes themselves, e.g., around $x = 1$ (Fig. 2B, lower panel). Our attribution measure $I_{local}(x)$ reflects this structure, peaking at $x = 0$, and decaying elsewhere (Fig. 2C). For the Gaussian prior, where posterior sensitivity is constant, $I_{local}(x)$ is flat. In contrast, previously proposed attribution methods behave inconsistently: some remain constant across both priors (Fig. 2D), others respond in the opposite direction (Fig. 2E), and some varied strongly even under a Gaussian prior, where the posterior sensitivity is uniform (Fig. 2F).

## 5.3 Data Processing Inequality

Next we illustrate how $I_{local}(x)$ respects the data processing inequality while certain other attribution methods do not. For this, we used a bimodal prior (Fig. 3A) and compared a linear-Gaussian neuron ($r \sim \mathcal{N}(x, \sigma_r^2)$) to a downstream neuron with response $r' = |r|$. Since this transformation is non-invertible, information must be lost. Consistent with the inequality, $I_{local}(x)$ decreased at every $x$ (Fig 3C). In contrast, both $I_{SSI}$ and $I_{sp}$ increased at some $x$ and decreased at others, violating the pointwise data processing inequality (Fi 3D-E). $I_{surp}$, by comparison, respected the inequality (Fig 3F).

## 5.4 Scaling to high-dimensions

We can use Eqn 20 to write the feature-wise decomposition in terms of the outputs of an unconditional and conditional diffusion model, trained to output $\hat{x}(x_\gamma) = E[X|x_\gamma]$ and $\hat{x}(x_\gamma, r) = E[X|x_\gamma, r]$,

respectively:

$$I_i(x) = \int_0^\infty \frac{1}{2\gamma^2} \mathbb{E}_{X_\gamma|x,R|X_\gamma} \left[ (\hat{x}_i(X_\gamma, R) - \hat{x}_i(X_\gamma))^2 \right] d\gamma \tag{21}$$

To obtain a Monte-carlo approximation of this expression, we need to sample from $x_\gamma \sim p(X_\gamma|x)$, followed by, $r \sim p(R|x_\gamma)$. Sampling from $p(R|x_\gamma)$ is prohibitively expensive (since it requires first sampling from $x \sim p(x_0|x_\gamma)$, which requires a full backward pass of the diffusion model). To get around this, we adopt an approximation used by Chung et al. 2023, instead approximating $I_i(x)$ using samples $r \sim p(R|\hat{x}(x))$, which can be computed efficiently using one pass through the de-noising network. The integral over $\gamma$ was approximated numerically with evenly spaced $\gamma$ (see Appendix C). Since the Fisher decays to zero for large $\gamma$, truncating the integral has little effect on our approximation of the integral.

### 5.5    Pixel-wise decomposition of encoded information

We applied our method to identify which regions of an image contribute most to the total information encoded by a population of visual neurons. For illustrative purposes, we used stimuli from the MNIST dataset and modelled a simple population of neurons with mean responses given by $Af(\mathbf{w} \cdot \mathbf{x} + b)$, where $A$ and $b$ are constants, $f(\cdot)$ is a sigmoid nonlinearity, and $\mathbf{w}$ is a linear filter representing the neuron's receptive field (RF). Neural responses were corrupted by Poisson noise. We simulated 49 neurons with RFs arranged in a uniform grid (Fig. 4A).

As a baseline, we first evaluated neural sensitivity using the diagonal of the Fisher information matrix (Fig. 4B–C, E–F). In this model, the Fisher information reduces to a weighted sum of squared RFs, where each neuron's contribution is scaled by its activation level. This yields characteristic "blob-like" patterns, with each blob centered on the neuron's RF.

We then used a diffusion model trained on MNIST to estimate the pixel-wise information decomposition, $I_{\text{local}}(\mathbf{x})$ (Fig 4D, G; additional images are shown in Supp Fig 2). This decomposition revealed that information was concentrated along object edges—regions where the decoded images (i.e. samples form the posterior) are most sensitive to small changes in the presented stimulus (cf. Fig. 2A). Unlike the Fisher information, our measure integrates how both the local sensitivity of neurons (via their RFs) and the statistical structure of the input (captured by the diffusion model), contribute towards the total encoded information.

Later, we investigated the behavior of our information decomposition on a diffusion model trained on natural images, with a model of recorded ganglion cell responses from the retina [15] (Appendix section C.6, and supplementary Figure 3). We observed qualitatively similar behavior to before, with $I_{local}$ peaking in regions of high local spatial contrast, around the edges of objects.

## 6    Discussion

We introduced a principled, information-theoretic measure of neural sensitivity to stimuli. This measure satisfies a core set of axioms that ensure interpretability and theoretical soundness. Crucially, the measure can be estimated using diffusion models, making it scalable to high-dimensional inputs and complex, non-linear neural populations. We empirically demonstrated how each axiom shapes interpretability through simple, illustrative examples. Finally, we show how the method can be applied in a high-dimensional setting to quantify the information encoded by a neural population about visual stimuli.

Kong et al. 2024 recently proposed two decompositions of the mutual information between visual stimuli $x$ and text prompts $y$, $I(x, y)$, which can be efficiently estimated using diffusion models. These were used to identify image regions most informative about accompanying text. Dewan et al. 2024 extended this approach to assess pixel-wise redundancy and synergy with respect to the prompt. However, these frameworks do not directly yield a stimulus-wise neural sensitivity measure $I(x)$, which was our goal (Appendix B). Moreover, simply averaging the decompositions of Kong et al. over neural responses does not produce stimulus-wise and feature-wise decompositions that satisfy our axioms: in one case the resulting decomposition is non-local and can be negative; in the other case, the decomposition is not additive, and thus violates a key information theoretic property pointwise.

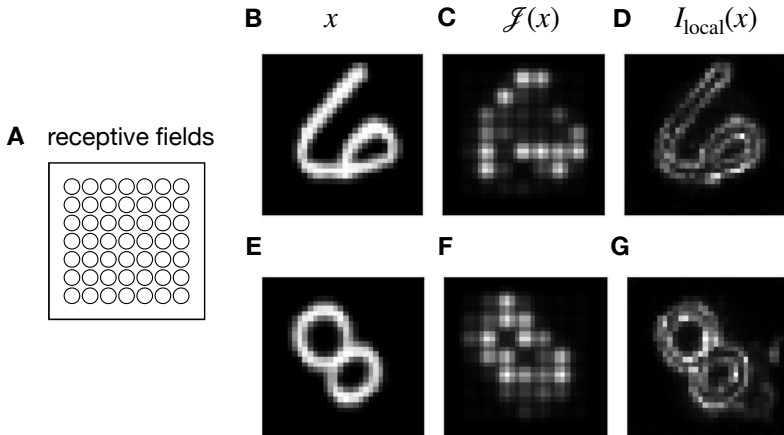

Figure 4: **Pixel-wise information decomposition with MNIST stimuli. (A)** Simulated population of linear–nonlinear–Poisson (LNP) neurons with circular receptive fields arranged in a grid. **(B)** Presented visual stimulus. **(C)** Diagonal of the Fisher information, given by a weighted sum of neural receptive fields. **(D)** Pixel-wise decomposition of mutual information, $I_{\mathrm{local}}(x)$. **E-G** Same as panels A-B, but for a different stimulus.

Our work extends a classical result from Brunel & Nadal 1998, who showed that mutual information can be approximated in the low-noise limit using Fisher information [29]. This approximation was later used by Wei & Stocker 2015 to explain a wide range of perceptual phenomena under the efficient coding hypothesis [21]. However, their approximation, which depends on the log determinant of the Fisher, becomes very inaccurate in certain cases, such as at high noise, or where there are more stimulus dimensions than neurons (in which case it returns minus infinity) [1, 31, 13]. Here, we instead identify an *exact* relation between mutual information and Fisher information with respect to a noise-corrupted stimulus. This opens new potential to test theories of efficient coding of high-dimensional stimuli, and in the presence of noise.

In the future, we will use our method to investigate more realistic neural models, fitted on biological data, as well as diffusion models trained on natural image datasets. Here, to further improve efficiency we could use zero-shot methods that sample directly from the posterior, without requiring a trained conditional diffusion model [22, 6]. Such approaches have recently achieved state-of-the-art performance in decoding visual scenes from retinal ganglion cell responses [30], and could enable rapid assessment of how changes to the neural model affect encoded stimulus information.

Finally, our axiomatic framework has close parallels with integrated gradients [26], a method developed to attribute the output of deep neural networks to individual input features. Both our method and integrated gradients address ill-posed attribution problems by enforcing natural axioms. However, one limitation of our method, in contrast to integrated gradients, is that we do not prove the uniqueness of our attribution method, as following from our axioms. This will be interesting to investigate in the future. There are also key differences between both approaches: for example, our attribution explicitly accounts for noisy responses and does not require specification of an arbitrary baseline. These distinctions make our method particularly well-suited to neural data, and suggest potential utility as a principled attribution tool for analyzing deep networks—identifying which stimuli, or stimulus features, different units or layers are sensitive to.

## Acknowledgements

Ulisse Ferrari and Simone Azeglio acknowledge this work was done within the framework of the PostGenAI@Paris project with the reference ANR-23-IACL-0007. Ulisse Ferrari and Simone Azeglio benefited from financial support by the Agence Nationale de la Recherche (ANR) by grants NatNetNoise (ANR-21 CE37-0024), IHU FOReSIGHT (ANR-18-IAHU-01). The PhD position of Simone Azeglio was funded by the Sorbonne Center for Artificial Intelligence (SCAI), through IDEX Sorbonne Université, project reference ANR-11-IDEX-0004. Matthew Chalk and Steeve Laquitaine

are funded by a grant RetNet4EC (ANR-22-CE92-0015), cofunded by the ANR and DFG. Our lab is part of the DIM C-BRAINS, funded by the Conseil Régional d'Ile-de-France.

We would also like to thank Tobias Kuhn and Olivier Rioul for interesting discussion and input, which helped us towards this paper.

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

# A Proof of locality axiom

Let $p(R, X)$ and $\tilde{p}(R, X)$ be two joint distributions, defined over the response, $R$, and a stimulus, $X$ with infinite support in an unbounded (e.g. $X \in \mathbb{R}^d$) or semi-bounded domain (e.g. $[0, \infty)^d$).

We restrict $\tilde{p}(R, X)$ to only differ from $p(R, X)$ locally, in the vicinity of $x_0$. Formally, we suppose there exists finite $\epsilon > 0$ and $x_0$ such that:

$$p(R, x) = \tilde{p}(R, x) \quad \text{for all } x \notin B_\epsilon(x_0), \tag{22}$$

where $B_\epsilon(x_0)$ is the open ball of radius $\epsilon$ centered at $x_0$.

Recall that our stimulus-wise decomposition of information is defined as:

$$I(x) = \frac{1}{2} \text{Trace} \int_0^\infty \mathbb{E}_{p(X_\gamma|x)} \left[ \mathcal{J}(X_\gamma) \right] d\gamma, \tag{23}$$

where $X_\gamma = x + \sqrt{\gamma} Z$, $Z \sim \mathcal{N}(0, I)$, and $\mathcal{J}(x_\gamma) = \mathbb{E}_{p(R|x_\gamma)} \left[ -\nabla^2_{x_\gamma} \log p(R \mid x_\gamma) \right]$ is the Fisher information matrix at $x_\gamma$.

We will consider the absolute difference between the stimulus-wise information, computed with $p(r, x)$ and $\tilde{p}(r, x)$ respectively, $|I(x) - \tilde{I}(x)|$. To prove the locality axiom, we separate the integral over $\gamma$ into two parts as follows,

$$|I(x) - \tilde{I}(x)| = \frac{1}{2} \left| \text{Trace} \int_0^\infty \mathbb{E}_{p(X_\gamma|x)} \left[ \mathcal{J}(X_\gamma) - \tilde{\mathcal{J}}(X_\gamma) \right] d\gamma \right| \tag{24}$$

$$\leq f(x, \gamma_h) + g(x, \gamma_h), \tag{25}$$

where

$$f(x, \gamma_h) = \frac{1}{2} \left| \text{Trace} \int_0^{\gamma_h} \mathbb{E}_{p(X_\gamma|x)} \left[ \mathcal{J}(X_\gamma) - \tilde{\mathcal{J}}(X_\gamma) \right] d\gamma \right| \tag{26}$$

$$g(x, \gamma_h) = \frac{1}{2} \left| \text{Trace} \int_{\gamma_h}^\infty \mathbb{E}_{p(X_\gamma|x)} \left[ \mathcal{J}(X_\gamma) - \tilde{\mathcal{J}}(X_\gamma) \right] d\gamma \right|. \tag{27}$$

In sections A.1, and A.2 we show that $f(x, \gamma_h)$ and $g(x, \gamma_h)$ have the following limiting behavior:

$$f(x, \gamma_h) = \mathcal{O} \left( e^{-\frac{1}{4\gamma_h} \|x - x_0\|^2} \right) \quad \text{as } \|x - x_0\|^2 \to \infty \tag{28}$$

$$g(x, \gamma_h) = \mathcal{O} \left( \frac{1}{\gamma_h} \right) \text{ uniformly over } x \text{ as } \gamma_h \to \infty. \tag{29}$$

Thus, for any $\delta > 0$, we can first pick $\gamma_h$ large enough so that $g(x, \gamma_h) \leq \delta/2$, for all $x$. Then, for fixed $\gamma_h$, we choose $d$ such that $f(x, \gamma_h) \leq \delta/2$ whenever $\|x - x_0\|^2 \geq d$.

This guarantees that for any $\delta > 0$, there exists a finite constant $d$ such that,

$$|I(x) - \tilde{I}(x)| \leq \delta \quad \text{for all } \|x - x_0\|^2 \geq d, \tag{30}$$

completing our proof of the locality axiom.

## A.1 Integral from $\gamma = 0$ to $\gamma = \gamma_h$

Here, we prove the limiting behavior of $f(x, \gamma_h)$, as $\|x - x_0\|^2 \to \infty$.

We start by writing out the Fisher information as a function of the noise-corrupted stimulus, $x_\gamma$:

$$\text{Trace}(\mathcal{J}(x_\gamma)) = \frac{1}{\gamma^2} E_{R|x_\gamma} \left( \|\hat{x}(x_\gamma) - \hat{x}(x_\gamma, r)\|^2 \right) \tag{31}$$

Writing out $\hat{x}(x_\gamma) \equiv E[X|x_\gamma]$ explicitly,

$$\hat{x}(x_\gamma) = \frac{\int x' p(x') \phi\left(\frac{x_\gamma - x'}{\sqrt{\gamma}}\right) dx'}{\int p(x') \phi\left(\frac{x_\gamma - x'}{\sqrt{\gamma}}\right) dx'} \tag{32}$$

$$= \frac{\int_{x' \notin B_\epsilon(x_0)} x' p(x') \phi\left(\frac{x' - x_\gamma}{\sqrt{\gamma}}\right) dx' + \int_{x' \in B_\epsilon(x_0)} x' p(x') \phi\left(\frac{x' - x_\gamma}{\sqrt{\gamma}}\right) dx'}{\int_{x' \notin B_\epsilon(x_0)} p(x') \phi\left(\frac{x' - x_\gamma}{\sqrt{\gamma}}\right) dx' + \int_{x' \in B_\epsilon(x_0)} p(x') \phi\left(\frac{x' - x_\gamma}{\sqrt{\gamma}}\right) dx'}, \tag{33}$$

where $\phi(x) \equiv \frac{1}{\sqrt{2\pi}} e^{-\frac{1}{2}|x|^2}$, and we have separated the integrals in the numerator and denominator into contributions from inside and outside the region $B_\epsilon(x_0)$. Taking just the second term in the numerator of Eqn 33,

$$
\begin{aligned}
\int_{x' \in B_\epsilon(x_0)} x' p(x') \, \phi\left(\frac{x' - x_\gamma}{\sqrt{\gamma}}\right) dx' \; &\leq \; \sup_{x' \in B_\epsilon(x_0)} \phi\left(\frac{x' - x_\gamma}{\sqrt{\gamma}}\right) \int_{x' \in B_\epsilon(x_0)} x' p(x') \, dx' \\
&= \; \phi\left(\frac{\|x_0 - x_\gamma\| - \epsilon}{\sqrt{\gamma}}\right) \int_{x' \in B_\epsilon(x_0)} x' p(x') \, dx' \\
&= \; \mathcal{O}\left(e^{-\frac{1}{2\gamma}\|x_0 - x_\gamma\|^2}\right) \quad \text{as} \quad \|x_0 - x_\gamma\|^2 \to \infty \quad (34)
\end{aligned}
$$

since, by construction, $p(x)$ has finite moments. The same logic also applies to the second term in the denominator of Eqn 33,

$$
\int_{x' \in B_\epsilon(x_0)} p(x') \, \phi\left(\frac{x' - x_\gamma}{\sqrt{\gamma}}\right) dx' = \mathcal{O}(e^{-\frac{1}{2\gamma}\|x_0 - x_\gamma\|^2}) \quad \text{as} \quad \|x_0 - x_\gamma\|^2 \to \infty. \quad (35)
$$

Substituting this asymptotic behaviour back into Eqn 33, we have

$$
\hat{x}(x_\gamma) = \frac{\int_{x' \notin B_\epsilon(x_0)} x' p(x') \, \phi\left(\frac{x' - x_\gamma}{\sqrt{\gamma}}\right) dx'}{\int_{x' \notin B_\epsilon(x_0)} p(x') \, \phi\left(\frac{x' - x_\gamma}{\sqrt{\gamma}}\right) dx'} + \mathcal{O}(e^{-\frac{1}{2\gamma}\|x_0 - x_\gamma\|^2}) \quad (36)
$$

where the first term only depends on $x$ in the region outside the region $B_\epsilon(x_0)$.

Using the same arguments for $\hat{x}(x_\gamma, r)$, we can write:

$$
\hat{x}(x_\gamma) - \hat{x}(x_\gamma, r) = a(x_\gamma, r, x_0) + \mathcal{O}(e^{-\frac{1}{2\gamma}\|x_0 - x_\gamma\|^2}), \quad (37)
$$

where

$$
a(x_\gamma, r, x_0) \equiv \frac{\int_{x' \notin B_\epsilon(x_0)} x' p(x') \, \phi\left(\frac{x' - x_\gamma}{\sqrt{\gamma}}\right) dx'}{\int_{x' \notin B_\epsilon(x_0)} p(x') \, \phi\left(\frac{x' - x_\gamma}{\sqrt{\gamma}}\right) dx'} - \frac{\int_{x' \notin B_\epsilon(x_0)} x' p(r, x') \, \phi\left(\frac{x' - x_\gamma}{\sqrt{\gamma}}\right) dx'}{\int_{x' \notin B_\epsilon(x_0)} p(r, x') \, \phi\left(\frac{x' - x_\gamma}{\sqrt{\gamma}}\right) dx'}, \quad (38)
$$

only depends on $p(r, x)$ in the region $x \notin B_\epsilon(x_0)$.

Taking the square, and averaging over $p(r|x_\gamma)$, we have

$$
\text{Trace}\left(\mathcal{J}(x_\gamma)\right) = \frac{1}{\gamma^2} E_{R|x_\gamma}\left[\|a(x_\gamma, r, x_0)\|^2\right] + \mathcal{O}(e^{-\frac{1}{2\gamma}\|x_0 - x_\gamma\|^2}) \quad (39)
$$

$$
= \frac{1}{\gamma^2} \frac{\int_{x, r} p(r|x) p(x) \phi\left(\frac{x - x_\gamma}{\sqrt{\gamma}}\right) \|a(x_\gamma, r, x_0)\|^2 dx}{\int_x p(x) \phi\left(\frac{x - x_\gamma}{\sqrt{\gamma}}\right) dx} + \mathcal{O}(e^{-\frac{1}{2\gamma}\|x_0 - x_\gamma\|^2}) \quad (40)
$$

Note that if $R$ is discrete, the above integral over $R$ is simply replaced by a sum. We then follow the exact same procedure as before, separating the integrals over $x$ into parts that are inside and outside of $B_\epsilon(x_0)$, to obtain

$$
\text{Trace}\left(\mathcal{J}(x_\gamma)\right) = \frac{1}{\gamma^2} \|b(x_\gamma, x_0)\|^2 + \mathcal{O}(e^{-\frac{1}{2\gamma}\|x_0 - x_\gamma\|}) \quad (41)
$$

where $b(x_0, x_\gamma)$ is a function that only depends on $p(r, x)$ in the region $x \notin B_\epsilon(x_0)$.

Since, by construction $\tilde{p}(r, x) = p(r, x)$ in the region $x \notin B_\epsilon(x_0)$, we can then write:

$$
\text{Trace}\left(\mathcal{J}(x_\gamma) - \tilde{\mathcal{J}}(x_\gamma)\right) = \mathcal{O}(e^{-\frac{1}{2\gamma}\|x_0 - x_\gamma\|}), \quad \text{as } \|x_0 - x_\gamma\|^2 \to \infty \quad (42)
$$

where $\tilde{\mathcal{J}}(x_\gamma)$ is the Fisher information obtained with $\tilde{p}(r, x)$.

Averaging over $p(X_\gamma|x) = \phi\left(\frac{X_\gamma - x}{\sqrt{\gamma}}\right)$ gives:

$$
\mathbb{E}_{p(X_\gamma|x)}\left[\text{Trace}(\mathcal{J}(x_\gamma) - \tilde{\mathcal{J}}(x_\gamma))\right] = \mathcal{O}\left(e^{-\frac{1}{4\gamma}\|x_0 - x\|^2}\right), \quad \text{as } \|x - x_0\|^2 \to \infty. \quad (43)
$$

Finally, integrating from $\gamma = 0$ to $\gamma_h$, we have:

$$f(x, \gamma_h) = \frac{1}{2}\left| \int_0^{\gamma_h} \text{Trace}\left(\mathbb{E}_{X_\gamma|x}\left[\mathcal{J}(X_\gamma) - \tilde{\mathcal{J}}(X_\gamma)\right]\right) d\gamma\right| \tag{44}$$

$$\leq \frac{1}{2}\int_0^{\gamma_h} \left|\text{Trace}\left(\mathbb{E}_{X_\gamma|x}\left[\mathcal{J}(X_\gamma) - \tilde{\mathcal{J}}(X_\gamma)\right]\right)\right| d\gamma \tag{45}$$

$$\leq \frac{1}{2}\gamma_h \sup_{\gamma \in (0, \gamma_h)} \left|\text{Trace}\left(\mathbb{E}_{X_\gamma|x}\left[\mathcal{J}(X_\gamma) - \tilde{\mathcal{J}}(X_\gamma)\right]\right)\right| \tag{46}$$

$$= \mathcal{O}\left(e^{-\frac{1}{4\gamma_h}\|x - x_0\|^2}\right) \quad \text{as } \|x - x_0\|^2 \to \infty, \tag{47}$$

as stated in Eqn 28.

## A.2 Integral from $\gamma = \gamma_h$ to $\infty$

Next, we show the limiting behaviour of $g(x, \gamma_h)$.

We know from Eqn 42 that for any given $\gamma < \infty$, $|\text{Trace}(\mathcal{J}(x_\gamma) - \tilde{\mathcal{J}}(x_\gamma))| \to 0$ as $|x_\gamma| \to \infty$. Thus the maximum of $|\text{Trace}(\mathcal{J}(x_\gamma) - \tilde{\mathcal{J}}(x_\gamma))|$ must be obtained for some finite $x_\gamma^*$. As a result

$$|\text{Trace}(\mathcal{J}(x_\gamma) - \tilde{\mathcal{J}}(x_\gamma))| \leq \sup_{x_\gamma} |\text{Trace}(\mathcal{J}(x_\gamma) - \tilde{\mathcal{J}}(x_\gamma))| \tag{48}$$

$$= |\text{Trace}(\mathcal{J}(x_\gamma^*) - \tilde{\mathcal{J}}(x_\gamma^*))|, \quad \text{where } |x_\gamma^*| < \infty \tag{49}$$

$$= \frac{1}{\gamma^2}\left| E_{p(r|x_\gamma^*)}\left(\|\hat{x}(x_\gamma^*) - \hat{x}\left(x_\gamma^*, r\right)\|^2\right)\right.$$

$$\left. - E_{\tilde{p}(r|x_\gamma^*)}\left(\|\tilde{x}(x_\gamma^*) - \tilde{x}\left(x_\gamma^*, r\right)\|^2\right)\right|, \quad \text{where } |x_\gamma^*| < \infty \tag{50}$$

with $\tilde{x}(x_\gamma)$ and $\tilde{x}(x_\gamma, r)$ defined as the expectations over $X$ obtained with the perturbed distribution, $\tilde{p}(R, X)$. Now since, by construction, $p(x)$, $p(x \mid r)$, $\tilde{p}(x)$, and $\tilde{p}(x \mid r)$ all have finite first and second moments, then for finite $x_\gamma^*$ all the expectations in the above expression are finite, and we have:

$$|\text{Trace}(\mathcal{J}(x_\gamma) - \tilde{\mathcal{J}}(x_\gamma))| \leq \frac{C}{\gamma^2}, \quad \text{for all } x_\gamma \tag{51}$$

where $C < \infty$ is some finite constant, independent of $x_\gamma$.

Integrating over $\gamma$ and averaging over $p(X_\gamma|x)$ gives

$$g(x, \gamma_h) = \frac{1}{2}\left|\text{Trace}\int_{\gamma_h}^\infty \mathbb{E}_{X_\gamma|x}\left[\mathcal{J}(X_\gamma) - \tilde{\mathcal{J}}(X_\gamma)\right] d\gamma\right| \tag{52}$$

$$\leq \frac{1}{2}\int_{\gamma_h}^\infty \left|\mathbb{E}_{X_\gamma|x}\left[\text{Trace}\left(\mathcal{J}(X_\gamma) - \tilde{\mathcal{J}}(X_\gamma)\right)\right]\right| d\gamma \tag{53}$$

$$\leq \frac{C}{2\gamma_h}, \quad \text{for all } x \tag{54}$$

$$= \mathcal{O}\left(\frac{1}{\gamma_h}\right), \quad \text{uniformly in } x, \tag{55}$$

as stated in Eqn 29.

## B Comparison with previous decompositions using diffusion models

Recently Kong et al. 2024 proposed two different decompositions of the mutual information into terms that depend on both the response and stimulus, which can be computed using diffusion models. Their decompositions take the following form:

$$I_{Kong,1}(x, r) = \int_0^\infty \frac{1}{2\gamma^2}E_{X_\gamma|x}\left[\|x - \hat{x}(x_\gamma, r)\|^2 - \|x - \hat{x}(x_\gamma)\|^2\right] d\gamma \tag{56}$$

$$I_{Kong,2}(x, r) = \int_0^\infty \frac{1}{2\gamma^2}E_{X_\gamma|x}\left[\|\hat{x}(x_\gamma) - \hat{x}(x_\gamma, r)\|^2\right] d\gamma \tag{57}$$

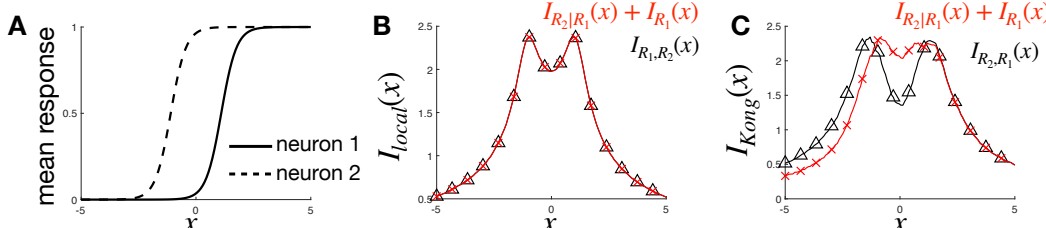

Figure 5: **Additivity, with respect to measurements from multiple neurons**. (A) We considered two neurons, with sigmoidal tuning curves, $f(x)$, and a 1-d stimulus with gaussian prior. (B) We confirmed that $I_{local}(x)$ satisfies additivity, by checking that the point-wise information from both neurons together ($I_{R,R_2}(x)$, red) equals $I_{R_1}(x) + I_{R_2|R1}(x)$. (C) For the measure derived from the work of Kong et al., which does not satisfy additivity, the curves are non-overlapping.

To obtain stimulus-wise decompositions from these expressions, that we can compare with our work, we averaged both of these decompositions with respect to $p(R|x)$, to obtain: $I_{Kong,1}(x) \equiv \mathbb{E}_{R|x}[I_{Kong,1}(x,R)]$ and $I_{Kong,2}(x) \equiv \mathbb{E}_{R|x}[I_{Kong,2}(x,R)]$.

Despite the apparent similarity with our proposed decomposition, neither $I_{Kong,1}(x)$ or $I_{Kong,2}(x)$ fulfill our axioms, limiting their potential use as principled & interpretable measures of neural sensitivity.

In their paper, Kong et al. show that $I_{Kong,1}(x,r) = \log \frac{p(r|x)}{p(r)}$. Taking the average over $p(R|x)$, gives: $I_{Kong,1}(x) \equiv \mathbb{E}_{R|x}[I_{Kong,1}(x,r)] = D_{KL}(p(R|x)\|p(R))$. This is identical to the expression for the specific surprise, $I_{surp}(x)$, in the main text (Eqn 3). As shown in the main text (Fig 1E), this measure does not fulfill our locality axiom. This is because $I_{surp}(x)$ depends on $p(r) = \int p(r|x')p(r')dx'$, which can be influenced by changes to the likelihood or prior with respect to any stimulus, $x'$. Further, following the same prescription as in the main text to obtain a feature-wise decomposition of $I_{Kong,1}(x)$, results in feature-wise attributions that can be negative.

To understand the second decomposition, we use Tweedie's law to write:

$$I_{Kong,2}(x) \equiv \mathbb{E}_{R|x}[I_{Kong,2}(x,r)] = \frac{1}{2}\int_0^\infty \mathbb{E}_{X_\gamma,R|x}\left[\|\nabla_{x_\gamma}\log p(R|X_\gamma)\|^2\right]d\gamma \qquad (58)$$

This differs from our expression, due to the fact that it involves taking the average over $p(R|x)$, rather than $p(R|x_\gamma)$. However, while this difference may seem subtle it has important consequences for the resulting stimulus-wise decomposition.

To see that there are large qualitative differences between $I_{Kong,2}(x)$ and $I_{local}(x)$ we first consider their behavior in the case where $p(R,X)$ is jointly gaussian. Here, $I_{local}(x)$ is constant across $x$ (Fig 2B, black), reflecting that the fact that the posterior is also independent of $x$ (up to a linear translation). In contrast, $I_{Kong,2}(x)$ scales quadratically with the distance of $x$ from the mean (similar to $I_{surp}(x)$, Fig 2E).

More importantly, $I_{Kong,2}(x)$ violates **additivity** point-wise. Supp Fig 1 illustrates this in a simple 1-d example. To see why it is the case, we can expand the expression for the local information from two neurons, $R$ and $R'$:

$$I_{R,R'}^{Kong,2}(x) = \frac{1}{2}\int_0^\infty \mathbb{E}_{X_\gamma,R,R'|x}\left[\|\nabla_{x_\gamma}\log p(R'|R,X_\gamma) + \nabla_{x_\gamma}\log p(R|X_\gamma)\|^2\right]d\gamma \qquad (59)$$

Now when we expand the square we see that the cross-terms *don't cancel out*. As a result, $I_{Kong,2}$ is not linear in $\log p(R'|R,X_\gamma)$ and $\log p(R|X_\gamma)$, violating additivity.

This contrasts with the behaviour of $I_{local}$, which can be expressed as follows:

$$I_{R,R'}^{local}(x) = -\frac{1}{2}\int_0^\infty \mathbb{E}_{X_\gamma|x},\left\{\mathbb{E}_{R',R|X_\gamma}\left[\nabla_{x_\gamma}^2\log p(R'|R,X_\gamma) + \nabla_{x_\gamma}^2\log p(R|X_\gamma)\right]\right\}d\gamma \qquad (60)$$

This equation is linear in $\log p(R'|R,X_\gamma)$ and $\log p(R|X_\gamma)$, and thus we can easily confirm that it is additive with respect to repeated measurements.

Additivity is a fundamental property that underpins the definition of mutual information [25]. It requires that information from multiple measurements (e.g., different neurons) combine linearly, such that their individual contributions sum to the total local information (Fig. 5). A point-wise measure that doesn't respect additivity could thus lead to misleading attributions as they don't respect how different measurements (or neurons) combine additively to generate the total mutual information.

## C  Diffusion model details

### C.1  Dataset Preparation

We utilized the MNIST dataset [19], which consists of 60,000 grayscale images of handwritten digits for training and 10,000 for testing. Each image, originally $28 \times 28$ pixels, was preprocessed by normalizing to the range $[-1, 1]$ and then rescaled to $32 \times 32$ pixels to align with the architectural requirements of our diffusion models.

### C.2  Neural Encoding Model

We simulated a population of 49 neurons with spatially localized receptive fields (RFs) arranged in a $7 \times 7$ grid, mimicking the retinotopic organization of early visual cortex. Image pixels were mapped to 2D coordinates in visual space, $(x, y) \in [-1, 1]^2$, forming a uniform grid. Each neuron's RF was modeled as a 2D Gaussian filter centered at $(x_i, y_i)$:

$$w_i(x, y) = \exp\left(-\frac{(x - x_i)^2 + (y - y_i)^2}{2\sigma^2}\right),\tag{61}$$

where $\sigma = 0.1$ defines the spatial extent of the RF, and $(x_i, y_i)$ specifies the center of the $i$-th neuron's RF. These centers were evenly spaced across the image grid.

Neural firing rates were computed by projecting the input image $I$ onto the receptive field weights $w_i$, followed by a sigmoid nonlinearity:

$$f_i = A \cdot \frac{1}{1 + \exp(g \cdot w_i^\top (I + 1 - \theta))},\tag{62}$$

where $A = 40$ is the amplitude, $g = 0.4$ is the gain and $\theta = 0.9$ is the threshold. This yields the mean firing rate of neuron $i$ in response to image $I$.

To capture neural variability, we modeled spike counts as samples from a Poisson distribution with mean $f_i$.

### C.3  Diffusion Models

We trained two denoising diffusion probabilistic models (DDPMs) based on the *UNet-2D* architecture [24], using the implementation provided by the HuggingFace `diffusers` library [27].

The first model was trained to generate MNIST digits from standard Gaussian noise $Z \sim \mathcal{N}(0, I)$. Following the DDPM framework [12], we simulated a forward diffusion process that progressively adds Gaussian noise to an image $x_0$, producing a sequence of noisy images $\{x_t\}$. The process is defined as:

$$x_t = \sqrt{1 - \beta_t} \cdot x_{t-1} + \sqrt{\beta_t} \cdot z_t, \quad z_t \sim \mathcal{N}(0, I),\tag{63}$$

where $\beta_t$ denotes the noise schedule. By recursively applying this equation, one obtains:

$$x_t \sim \mathcal{N}\left(\sqrt{\bar{\alpha}_t} \cdot x_0, (1 - \bar{\alpha}_t)I\right), \quad \text{with} \quad \bar{\alpha}_t = \prod_{s=1}^{t}(1 - \beta_s).\tag{64}$$

Although this differs from the isotropic Gaussian noise model used in the main text, $x_t = x_0 + \sqrt{\gamma}Z$, the two formulations are equivalent up to reparameterization, with $\gamma = \frac{1 - \bar{\alpha}_t}{\bar{\alpha}_t}$. Thus, both can be used interchangeably to estimate the information decomposition $I_{\text{local}}(x)$.

We used a linear noise schedule with $\beta_t$ increasing from $1 \times 10^{-4}$ to $0.02$ over $1{,}000$ time steps. Given a noisy image $x_t$, the model was trained to predict the noise component $\hat{z}_t(x_t)$, which can be used to reconstruct an estimate of the clean image:

$$\hat{x}(x_t) = \frac{x_t - \sqrt{1 - \bar{\alpha}_t} \cdot \hat{z}_t(x_t)}{\sqrt{\bar{\alpha}_t}}. \tag{65}$$

A second DDPM was trained using the same architecture, but conditioned on simulated neural responses $r$ (see previous section). This model learned to predict the noise given both the noisy image and response:

$$\hat{x}(x_t, r) = \frac{x_t - \sqrt{1 - \bar{\alpha}_t} \cdot \hat{z}_t(x_t, r)}{\sqrt{\bar{\alpha}_t}}. \tag{66}$$

We used these models to estimate the pixel-wise information decomposition described in the main text (Eq. 9). To approximate the integral over $\gamma$, we applied additive Gaussian noise according to the diffusion schedule and evaluated $\hat{x}(x_t)$ and $\hat{x}(x_t, r)$ over 24 diffusion steps (indexed as $[40{:}40{:}1000]$) using 50 noisy samples of $x_t$ and $r$ at each $t$. On average, approximating $I_{local}(x)$ for one image took approximately 1 minute on an NVIDIA GeForce RTX 4080 GPU.

## C.4  Training Details

Both models were implemented in PyTorch (version 2.7.0) with Cuda 12.6 and trained using the HuggingFace Diffusers library [27] and Accelerate library [11] for distributed training on 7 NVIDIA A100 GPUs (40 GB VRAM).

The training procedure for both models was as follows:

- Optimizer: AdamW with learning rate $1 \times 10^{-4}$, $\beta_1 = 0.9$, $\beta_2 = 0.999$
- Learning rate scheduler: Cosine schedule with warmup (500 steps)
- Batch size: 256
- Training duration: 150 epochs
- Loss function: Mean squared error (MSE) between predicted and true noise
- Precision: Mixed precision training with bfloat16
- Training Time: 25/30 minutes per model

## C.5  Decomposition for additional digits

We repeated the analysis described in Figure 4 originally performed for two digits on six additional digits using exactly the same linear-nonlinear Poisson (LNP) model, receptive field configuration, and information decomposition procedure. For each new digit, we computed the Fisher information and the pixel-wise information $I_{local}(x)$ following identical preprocessing, stimulus presentation, and estimation steps as in the main analysis.

## C.6  Decomposition for natural images

To demonstrate that our approach generalizes to more complex and biologically realistic neural encoder models as well as natural images, we trained the diffusion models on 60,000 64×64 natural images from Hugging Face's Tiny ImageNet dataset and trained a deep neural network model of the retina [15] to replicate the responses of 41 retinal ganglion cells (RGCs) from [10]. We extended the model by tiling the stimulus space with a 7×7 square lattice mosaic of receptive fields, where the resulting 49-field mosaic reproduced the response pattern of one of the fitted cells. As with the MNIST digits, we then applied pixel-wise information decomposition to natural images (Fig. 7). Consistent with our findings for the MNIST stimuli, the resulting local information maps, $I_{local}(x)$, exhibited the highest values in regions of high spatial contrast, particularly along object boundaries, indicating that these areas contribute most strongly to the encoded information about the stimulus.

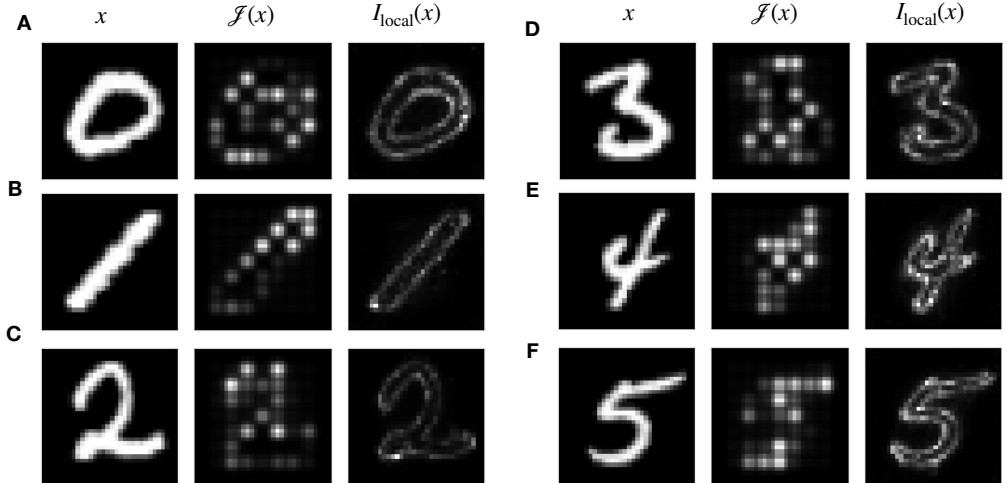

Figure 6: **Information decomposition across additional digits.** In the main text, we present pixel-wise information decomposition for two example digits. Here, we provide several additional examples to illustrate the decomposition patterns across other digits.

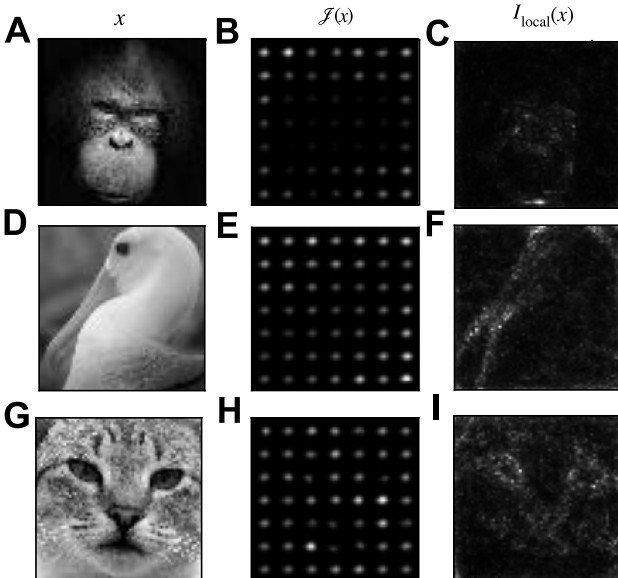

Figure 7: **Pixel-wise information decomposition with natural image stimuli. (A)** Presented visual stimulus. **(B)** Fisher information was computed as the sum of each neuron $i$'s pointwise contribution, $(f_i'(x))^2/f_i(x)$, quantifying the local neural sensitivity to infinitesimal changes in the intensity of pixel $x$. **(C)** Pixel-wise decomposition of mutual information, $I_{\mathrm{local}}(x)$. **D-I** Same as panels A-B, but for different stimuli.

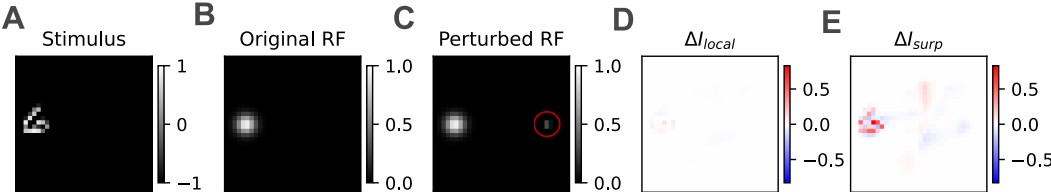

Figure 8: **Illustration of the locality of $I_{\mathrm{local}}(x)$ compared to $I_{\mathrm{surp}}(x)$. (A)** The visual stimulus is a MNIST digit positioned on the center-left side of the visual field. **(B)** Receptive field of LNP model neuron 21 in the unperturbed condition, located on the center-left side of the visual field (the remaining 48 receptive fields are unperturbed and not shown). **(C)** The receptive field of neuron 21 is modified by adding a sharp 2D Gaussian perturbation (highlighted by a red circle) on the opposite side of the stimulus. **(D)** Difference in $I_{\mathrm{local}}(x)$ and **(E)** difference in $I_{\mathrm{surp}}(x)$ between the original and perturbed receptive field conditions. $I_{\mathrm{local}}(x)$ remained confined near the stimulus and the perturbed receptive field, reflecting its local nature, while $I_{\mathrm{surp}}(x)$ exhibited broader, spatially distributed changes.

## C.7 Demonstration of locality for MNIST stimuli

To further illustrate the locality properties of the information decomposition, we examined how a small, spatially localized perturbation to a single receptive field affects $I_{\mathrm{local}}(x)$ and $I_{\mathrm{surp}}(x)$. We used the same linear–nonlinear–Poisson (LNP) model described in the main text, with receptive fields arranged across the visual field (Supplementary Figure 8B). In the perturbed condition, we introduced a sharp two-dimensional Gaussian bump to the receptive field of a single neuron, located on the opposite side of the visual stimulus (Figure 8C).

We then computed the pixel-wise differences in $I_{\mathrm{local}}(x)$ and $I_{\mathrm{surp}}(x)$ between the perturbed and unperturbed conditions. As shown in Figure 8D–E, $I_{\mathrm{local}}(x)$ remained confined to the vicinity of the stimulus and the affected receptive field, reflecting its inherently local nature. In contrast, $I_{\mathrm{surp}}(x)$ exhibited broader, spatially distributed changes, highlighting its global dependence on the overall structure of the neural population code.

The code repository to reproduce figure 4, 6, 7 and 8 can be found at neural-info-decomp.

