# OpenReview forum: "Decomposing stimulus-specific sensory neural information via diffusion models"
_NeurIPS.cc/2025/Conference — NeurIPS 2025 spotlight_

### Official Review · Reviewer_2L2J · 2025-06-16

**Clarity:** 3
**Significance:** 2
**Originality:** 3
**Rating:** 4
**Confidence:** 3

**Summary:**

This paper proposes a new measure of neural sensitivity $I_{local}(x)$ that satisfies certain desired properties of completeness, locality, positivity, additivity, and insensitivity to irrelevant features. The authors use de-Brujin's identity to derive a decomposition of mutual information between response and stimuli, $I(R;X)$, into a stimulus-specific quantity $I_{local}(x)$. The proposed $I_{local}(x)$ is essentially a conditionally-averaged non-parametric Fisher information of a gaussian-noise corrupted version of the stimuli. The authors claim that their proposed decomposition satisfies all the desired properties whereas all other existing decompositions don't, particularly, the locality axiom. Furthermore, authors show that their proposed decomposition can be easily computed using a conditional and an unconditional DDPM models. The authors compare their decomposition with existing decompositions in a toy example and simulated visual neurons with MNIST dataset. The authors conclude by arguing that their decomposition is more interpretable.

**Questions:**

1. Why other decomposition methods were not used to study the decomposition in the MNIST dataset?

**Ethical Concerns:**

["NO or VERY MINOR ethics concerns only"]

**Final Justification:**

I have updated my score based on the author's rebuttal. Author's were able to successfully answer my most pressing concerns.

**Limitations:**

Yes

**Quality:**

1

**Strengths And Weaknesses:**

*I have provisionally rejected based on the technical flaws listed in the weakness of the paper section. if authors are able to satisfactorily address all my concerns below, I am happy to raise my recommendation to accept.*

**Strength of the paper**:

1. The paper is well-written and properly formatted.
2. The paper attempts to follow a principled axiomatic approach to derive a measure of information, ensuring that the derived measure is theoretically grounded and interpretable.
3. The paper does a fairly good job of exposing previous literature.
4. I especially enjoyed the inclusion of intuitive text/english descriptions accompanying the math.
5. The simulation results are illustrating although the results would have been more stronger if there was real data. Any particular reasons why Allen brain visual dataset was not used (see Venkatesh et al, NeurIPS'23)?
6. The paper provides sufficient details to understand the computational experiments and does a good job of discussing its limitations.



**Weaknesses of the paper**: The main weakness of the paper is the lack of rigor in the mathematical/theoretical analysis. A lot of theoretical statements are made heuristically and w/o assumptions. A simple example is that authors use Fisher information to define their decomposition, but do not state under what assumptions (e.g., smoothness, continuity on the pdfs) the Fisher information is well-defined. Fisher information requires fairly strong regularity conditions (such as existence of a second derivative) to be well-defined. Such assumptions should be clearly mentioned somewhere. I understand in practice such assumptions are typically satisfied as we work with nice distributions. Regardless, it is important to mention these assumptions as the main content of this paper is theoretically-focused.

I briefly state the most concerning issues with the theoretical analysis below:

1. **Use of de-Brujin's identity to conditional differential entropy**:

Using author's notation, we can summarize the vanilla version of the de-Brujin's identity as follows:

$$ \frac{dh(X+\sqrt{\gamma}Z)}{d\gamma} = \frac{dh(X_{\gamma})}{d\gamma}  =\frac{1}{2}J(X_{\gamma})= -\frac{1}{2}\text{Trace}\left[\mathbb{E}\left[\nabla^2\log\left(p(X_{\gamma})\right)\right]\right]; Z\sim\mathcal{N}(\mathbf{0},\mathbf{I}), J(\cdot)\text{ is the Fisher information}.$$

The above equation (i.e., the vanilla de-Brujin's identity) only relates the differential entropy to the non-parametric Fisher information using a Gaussian channel. It **does not** relate the conditional differential entropy $h(X_{\gamma}|R)$ to the Fisher information. Note that the conditional differential entropy and differential entropy are different mathematical entities. To illustrate that, we can explicitly write out the formulas for $h(X_{\gamma})$ and $h(X_{\gamma}|R)$ (assuming the underlying pdfs satisfy the regularity conditions):

$$h(X_{\gamma}) = -\int p(X_{\gamma})\log(p(X_{\gamma}))dX_{\gamma}\text{  and  }h(X_{\gamma}|R) = -\int p(R)p(X_{\gamma}|R)\int\log(p(X_{\gamma}|R))dX_{\gamma}dR.$$

Note that the term inside and outside of the $\log(\cdot)$ is the same in the integrand of the $h(X_{\gamma})$. This is not the case in $h(X_{\gamma}|R)$ (note the extra $p(R)$). Hence, I am skeptical the vanilla version of de-Brujin's identity can be applied to $h(X_{\gamma}|R)$, as is done in equation (5) of the paper. To elaborate, let us analyze the term $\frac{dh(X_{\gamma}|R)}{d\gamma}$. Assuming the underlying pdfs satisfy all the required regularity condition, we can expand the term as follows:

$$
 \frac{dh(X_{\gamma}|R)}{d\gamma} =- \frac{d}{d\gamma}\int p(R)\int p(X_{\gamma}|R)\log(p(X_{\gamma}|R))dX_{\gamma}dR
$$
Using chain rule and exchanging the differential and integration operators:
$$
 \frac{dh(X_{\gamma}|R)}{d\gamma} =- \int \frac{dp(R)}{d\gamma}\left[\int p(X_{\gamma}|R)\log(p(X_{\gamma}|R))dX_{\gamma}\right]dR-\int p(R) \underbrace{\frac{d\left[\int p(X_{\gamma}|R)\log(p(X_{\gamma}|R))dX_{\gamma}\right]}{d\gamma}}_{\text{term 1}}dR
$$

Now, it may be possible to apply de-Brujin's identity to "term 1" as the  terms inside and outside of $\log(\cdot)$, i.e. $p(X_{\gamma}|R)$, are the same. Technically, one would also need to show that Z and X are conditionally independent given R, (*which they may not be*) to apply de-Brujin's identity to term1. Assuming the (best case scenario) that  Z and X are conditionally independent given R, we can apply de-Brujin's identity in the previous equation to obtain:

$$
 \frac{dh(X_{\gamma}|R)}{d\gamma} =- \int \frac{dp(R)}{d\gamma}\left[\int p(X_{\gamma}|R)\log(p(X_{\gamma}|R))dX_{\gamma}\right]dR+\frac{1}{2}\text{Trace}\left[ \mathbb{E}\left[\nabla^2_{x_{\gamma}}p(X_{\gamma}|R)\right]\right]
$$

For equation (5) in the manuscript to be correct, the authors need to show that first term on the RHS of the above equation is **zero**. Based on my reading of the manuscript, it seems the underlying (natural) assumption is $p(R|X=x)=p(R|X_{\gamma}=x)$, as $X_{\gamma}$ is thought of as a noisy stimulus being provided to the system (Please correct me if my assumption is wrong). Hence, $\frac{dp(R)}{d\gamma}$ would not be zero as $p(R)=\int p(R|X_{\gamma}=x)p(X_{\gamma}=x)dx$ should depend on $\gamma$.

I would recommend authors:

i. To either point me to a citation that shows a version of de-Brujin's identity that explicitly relates the conditional differential entropy and Fisher information (I skimmed through Section D. of [Rioul, 2010] and could not find it), or

ii. To rigorously show that vanilla de-Brujin's identity can be applied to conditional differential entropy, with all the assumptions and proof steps clearly stated.

Note that if the application of de-Brujin's identity on conditional differential entropy cannot be rigorously proven, then the proposed $I_{local}(x)$ does not satisfy the completeness axiom, and leaves the manuscript on shaky ground.

2. **Lack of mathematical rigor in defining Axiom 2: Locality**

i. My first issue is with the equation $p(R,x)=\tilde{p}(R,x)$ for all $x\notin B_{\epsilon}(x_0)$. This is too strong a statement. The perturbations are not constrained and can be anything inside the $B_{\epsilon}(x_0)$. This is problematic, as I can define $\tilde{p}(R,x)$ to be a distribution that is, e.g. not differentiable, in the region contained in $B_{\epsilon}(x_0)$. Consequently, the Fisher information for $\tilde{p}(R,x)$ would not be defined in $B_{\epsilon}(x_0)$, making the decomposition $I_{local}(x)$ undefined for $\tilde{p}(R,x)$. For example, choose $X$ and $R$ to be scalar, and $\epsilon$ to be 1. Define $\tilde{p}(X|R) = A(R)|W(x)|$ for $|X|\leq 1$ and $\tilde{p}(X|R) = p(X|R)$ for $|X|\geq 1$, where $A(R)$ is a normalizing factor to ensure the probability distribution  integrates to one and W(x) is the Weierstrass function. Furthermore, keep $p(R)=\tilde{p}(R)$. Now, $\tilde{p}(X|R)$ is not differentiable for $|x|\leq 1$ and hence the term $\nabla^2\log(\tilde{p}(X|R))$ would be undefined for $|x|\leq 1$. Therefore, we are unable to compute Fisher information and $I_{local}(x)$ for $\tilde{p}(X,R)$. This example demonstrates the need for the perturbations to be constrained as one can create **pathological examples** to show that $I_{local}(x)$ may not be meaningfully *defined* much less satisfy the locality axiom.

ii.  The limiting procedure, i.e., $|I(x)-\tilde{I}(x)|\rightarrow 0$ when $\|x-x_0\|\gg d$ is awkwardly phrased. Typically, the way to propose a limiting procedure is to show an "input" ($x$ moving along a certain direction while the output ($|I(x)-\tilde{I}(x)|$) is converging to a quantity. The condition $\|x-x_0\|\gg d$ does not show a movement of the input $x$, which makes the limiting operation a bit hard to interpet. I would encourage authors to use the more conventional way of defining limits. For example, for every $\delta>0$ there exists a $d_{\delta}$ such that $|I(x)-\tilde{I}(x)|\leq \delta$ when $\|x-x_0\|\geq d_{\delta}$.

iii. The proposed Axiom 2 is problematic for finite domains. For example, let us assume the domain of $X$ is finite, then I can always choose an $\epsilon$ big enough such that $B_{\epsilon}(x_0)$ contains the whole domain of $X$. Hence, I can choose $\tilde{p}(R,X)$ that is completely different than $p(R,X)$. Consequently, no possible decomposition could ever satisfy the Locality Axiom, as I could always find $\tilde{p}(R,X)$ (for example by adding noise everywhere) such that $|I(x)-\tilde{I}(x)|>\epsilon\ \forall x$.

Axioms are the heart of any mathematical theory. It is important to ensure that the axioms are defined and written properly such that the pathological examples (as above) do not occur. I would encourage authors to re-think and re-work their Axiom 2.

3. **Lack of rigor in the Proof presented in Appendix A**.

i. My first concern is from equation (11)-(13). I am interpreting their statement (above equation (11)) as $\gamma\geq d$, where $d$ is large (it is a bit confusingly written: "large $\gamma>d$"). Now it seems that authors are arguing in equation (12) that $E[X|x_{\gamma, R}]\approx E[X|R]$ amd $E[X|x_{\gamma}]\approx E[X]$, but provide no reasoning as to why it should be true. I am assuming that the rationale is that $X_{\gamma}$ for large $\gamma$ is just noise and consequently independent from $X$. There are concerns with that line of reasoning:

a. $X$ being approximately independent of $X_{\gamma}$ does not necessarily imply $X$ is approximately **conditionally independent**  of $X_{\gamma}$. The approximate conditional independence needs to be rigorously shown by authors. Please look at the famous XOR example to gain intuition regarding the difference between conditional independence and independence.

b. The change of $\gamma^2$ to $d^2$ from equation (11) to equation (12) is abrupt and without explanation.

c. Please do not use $\approx$ in a proof. Please be precise in your statements. For example, use statements such as $\mathcal{J}(x_{\gamma}) = \frac{1}{d^2}\Sigma_{X|R}+\text{error term}$, where the error term goes to zero as $\gamma$ goes to infinity.

d. Assuming equation (12) is true, i.e., $\mathcal{J}(x_{\gamma}) \approx \frac{1}{d^2}\Sigma_{X|R}$, the integral $\text{Trace}[\int_{d}^{\infty} E_{X_{\gamma}|x}[J(x_{\gamma})] d\gamma]$ can be simplified as follows:

$$\text{Trace}[\int_{d}^{\infty} E_{X_{\gamma}|x}[J(x_{\gamma})] d\gamma] \approx \frac{1}{d}\text{Trace}[ \int_{d}^{\infty} E_{X_{\gamma}|x}[\Sigma_{X|R}]d\gamma] = \frac{1}{d}\int_{d}^{\infty} \text{Trace}[\Sigma_{X|R}]d\gamma,$$

where I substituted the approximation proposed in equation (12) followed by the fact that $\text{Trace}[\Sigma_{X|R}]$ does not depend on $X_{\gamma}$ and hence we can remove the expectation operator. There has to be some error in this approximation as $ \frac{1}{d}\int_{d}^{\infty} \text{Trace}[\Sigma_{X|R}]d\gamma$ for any positive posterior covariance ( $\text{Trace}[\Sigma_{X|R}]$) would be infinite and not $\frac{1}{d}\text{Trace}[\Sigma_{X|R}]$ as stated in equation (13).

e.  In equation (12), authors are equating a *scalar* with a *vector*:  $\|\| E[X|R]-E[X]\|\|^2= \Sigma_{X|R}$. I guess the authors meant $\|\| E[X|R]-E[X]\|\|^2= Trace[\Sigma_{X|R}]$? There is also a typo regarding the factor of $\frac{1}{2}$, the factor of $\frac{1}{2}$ should appear in equation (13).

ii. Statement, "This is a convolution with Gaussian, with width $\gamma$ ... vanishing influence on $\log(p(R|x_{\gamma}))$". Don't state but prove this statement either through a citation or a proof. The goal is indeed to show that this statement is true.

iii. Equation (15) provides an approximation for $\|\| x_{\gamma}-x_{0}\|\|^2\gg \gamma$ whereas eqns (11)-(13) provide an approximation for $\gamma>d$. Why is the analysis of the region between $\|\| x_{\gamma}-x_{0}\|\|^2\gg \gamma$ and $\gamma>d$, i.e., when $\sqrt(\gamma)$ and $d$ are roughly of the same order, **omitted**. Note that the integration goes from 0 to $\infty$. Hence, an approximation for that region is also needed.

iv There is a typo, it should be $\gamma$ in equation (14).

I would encourage authors to clearly define the approximations for each segment of the integral and then show that $I_{local}(x)$ satisfies some desired locality properties.

**Minor comments**

i. Please do not change the order of intergrals, differential, and limiting operations without stating the appropriate regularity conditions. For example, to show that $E_{X}[I_{local}(x)] = I(R;X)$, you need dominated convergence theorem to move the expectation operator $(E_{X})$ inside the integral in the definition of $I_{local}(x)$ (provided in equation (8)) to obtain the term $E_{X}[\mathcal{J}_{ii}(X\gamma)]$ . Note that it not always possible to change the order of intergals, differential operators when infinite limits are involved.

ii. Please do not confuse Entropy with differential entropy. Differential entropy is denoted by $h(\cdot)$ whereas the entropy function is denoted by $H(\cdot)$. Differential entropy and entropy are different mathematical entities which is underscored by the fact that differential entropy can be negative whereas entropy is always positive.

iii. Please clearly state assumptions on the pdf that allow application of de-Brujin's identity, existence of differential entropy $h(X)$. For example, consult [Rioul, 2010] for the required assumptions.

---

> ### Author Rebuttal · Authors · 2025-07-28
>
> We thank reviewer 2L2J for their detailed & insightful review. We agree that the rigor of some proofs and mathematical statements can be improved, and we appreciate the reviewer’s comments, which will help us strengthen the final version. In particular, we acknowledge that Appendix A was not sufficiently clear, and we thank the reviewer for their comments, which will help us improve it. That said, the key results presented in the paper are correct, as clarified below. We hope our responses address the reviewer’s concerns.
>
> =============== Comment 1 ==============
>
> We believe the confusion arises from a misunderstanding of the model: specifically, the reviewer assumes that $p(R|X=x) = p(R| X_\gamma=x)$, which is not correct in our setting. We thank the reviewer for highlighting this point and allowing us to clarify.
>
> The response $R$ is generated from the original stimulus $X$, \& not from $X_\gamma$, according to $p(R|X)$. The noise corrupted stimulus, $X_\gamma$, equals the stimulus $X$, plus gaussian noise, $Z$. Thus, $R$ and $X_\gamma$ are conditionally independent given $X$, and the joint distribution factorizes as $$p(R,X,X_\gamma)=p(R | X)p(X_\gamma | X)p(X).$$
>
> Marginalizing over $X$ and $X_\gamma$, we find that $p(R)$ is independent of $\gamma$, i.e.,
> $$\frac{d}{d\gamma}p(R)=0.$$
> As a result, the first term in the reviewer’s derivation vanishes.
>
> We are left with what the reviewer calls `term 1' which includes: $$\frac{d h(X+\sqrt{\gamma}Z|R=r)}{d\gamma}=-\frac{d}{d\gamma}\int p(X+\sqrt{\gamma}Z |R=r)\log p(X+\sqrt{\gamma} Z|R=r)dXdZ.$$ Now, as (from above) $R-X-X_\gamma$ forms a Markov chain, we can write $p(X_\gamma,X|r)=p(X_\gamma|X)p(X|r)$, which also implies, $p(Z, X|r)=p(Z|X)p(X|r)$. Since, by construction, $p(Z|X)=p(Z)$, we establish that $Z$ is also *conditionally* independent of $X$ given $r$, as required for de-Brujin's identity.
>
> =============== Comment 2 ==============
>
> i. We can exclude the pathological example raised by the reviewer by stating that both $p(R,X)$ and $\tilde p(R,X)$ satisfy standard regularity conditions for Fisher Information. Specifically, we will state that $p(R,X)$ and $\tilde p(R,X)$ are doubly differentiable for all $X$, with sufficient decay at infinity for $\mathcal{J}(x)$ to exist and be finite everywhere (Rioul 2010). We will add this statement to the camera-ready version.
>
> ii. We agree, and we will present the limiting procedure more rigorously in the revised version, as suggested by the reviewer: namely, that for every $\delta>0$ there exists a $d_\delta$ such that $|I(x)-\tilde I(x)|\leq \delta $ when $||x-x_0||^2\geq d_\delta$.
>
> iii. We assume in the paper that $x\in\mathbb{R}^d$, and thus has infinite support (line 68). We will clarify in revision.  This is a reasonable assumption for most distributions of interest in sensory neuroscience. However, we will add a sentence to the discussion to state this limitation, and that future work could deal with bounded variables.
>
> =============== Comment 3  ==============
>
> ia. The reviewer is correct in stating that we reason that, in the high-$\gamma$ limit, $X$ becomes independent of $X_\gamma$, since $X_\gamma$ is just noise. Moreover, for our model, where $R$ is conditionally independent of $X_\gamma$ given $X$, the joint factorises as $p(X, R, X_\gamma) = p(R|X)p(X_\gamma|X)p(X)$ (See comment 1). Thus, when $X_\gamma$ becomes independent of $X$ (so that $p(X_\gamma|X)=p(X_\gamma)$), $X$ also becomes *conditionally independent* of $X_\gamma$, given $R$.
>
> ib-e. Agreed, we mistakenly replaced $\gamma$ with $d$ in Eqn 12. Instead we should state that, in the high-$\gamma$ limit,$$\mathcal{J}(x_\gamma) = \frac{1}{\gamma^2} \Sigma_{X|R} + o\left(\frac{1}{\gamma^2}\right)$$. Thus, the integration becomes,$$\int_{d}^{\infty}\frac{1}{\gamma^2}\mathrm{Tr}(\Sigma_{X|R})d\gamma =\frac{1}{d}\mathrm{Tr}(\Sigma_{X|R}),$$which is the result stated in the appendix.
>
> ii. We agree this statement needs proved. We will add the following to the appendix.
>
> Consider Equation 14 from the paper:
> $$\log p(r|x_\gamma)=\log \frac{\int_{x'}p(r, x')\phi\left( \frac{x'-x_\gamma}{\sqrt{\gamma}}\right)dx'}{\int_{x'}p(x')\phi\left( \frac{x_\gamma-x'}{\sqrt{\gamma}} \right)dx'},$$
> where $\phi(\cdot)$ denotes the standard multivariate normal density. We aim to show that $\log p(r \mid x_\gamma)$ is unaffected by perturbations to $p(r,x)$ that are localized within the domain $x \in B_\epsilon(x_0)$, in the limit where $||x_\gamma-x_0||^2 \gg \gamma$.
>
> For this, we split the integrals (numerator and denominator) into 2 regions: $x'\notin B_\epsilon(x_0)$ (where $p(r,x')=\tilde{p}(r, x')$), \& $x' \in B_\epsilon(x_0)$ (where $p(r,x') \ne \tilde{p}(r,x')$).
>
> Then:
> $$\log p(r|x_\gamma) = \log \frac{\int_{x'\notin B_\epsilon(x_0)} p(r, x')\phi\left( \frac{x_\gamma - x'}{\sqrt{\gamma}} \right) dx'+\int_{x' \in B_\epsilon(x_0)}p(r, x')\phi\left(\frac{x_\gamma - x'}{\sqrt{\gamma}} \right) dx'
> }{\int_{x' \notin B_\epsilon(x_0)}p(x')\phi\left(\frac{x_\gamma -x'}{\sqrt{\gamma}}\right)dx'+\int_{x'\in B_\epsilon(x_0)}p(x')\phi\left( \frac{x_\gamma -x'}{\sqrt{\gamma}} \right)dx'}$$
>
> The first terms in both numerator and denominator are independent of changes to $p(r,x)$ within $B_\epsilon(x_0)$. To complete the argument, we thus need to show that the second terms tend to zero as $||x_\gamma - x_0||^2 \gg \gamma$.
>
> If $p(r,x)$ is locally bounded in $B_\epsilon(x_0)$ (which follows from the regularity assumptions stated in comment 2i, that the distribution is both integrable and differentiable everywhere), i.e., there exists $M< \infty$ such that
> $$\sup_{x'\in B_\epsilon(x_0)}p(r, x')\leq M.$$
> Then:
> $$\int_{x'\in B_\epsilon(x_0)} p(r, x')\phi\left( \frac{x_\gamma-x'}{\sqrt{\gamma}} \right) dx'
> \leq M\int_{x' \in B_\epsilon(x_0)}\phi\left( \frac{x_\gamma-x'}{\sqrt{\gamma}} \right) dx'.$$
>
> The right-hand side is the mass of a Gaussian centered at $x_\gamma$ over a ball that is far from its mean—this decays rapidly as $||x_\gamma-x_0||^2\gg \gamma$. Therefore, both the numerator and denominator correction terms vanish in this limit.
>
> This shows that $\log p(r|x_\gamma)$ becomes insensitive to local perturbations in $p(r,x)$ far from $x_\gamma$, as claimed.
>
> iii. We thank the reviewer for highlighting this point—we agree that our original explanation lacked sufficient clarity. However, while we acknowledge the proof could be made more rigorous, we maintain that our claimed result is true. For the camera ready version, the proof can be made significantly more rigorous, with minor modifications to the mathematical statements and notation, as set out below.
>
> Recall that goal is to show that
> $$I_{local}(x)=\frac{1}{2}\int_{0}^{\infty}E_{X_\gamma |x}[ \mathcal{J}(X_\gamma)]d\gamma$$
> is unaffected by perturbations to $p(r, x')$ restricted to the region $x' \in B_\epsilon(x_0)$, provided that $||x-x_0||^2$ is sufficiently large (the locality axiom; see comment 2ii).
>
> To show this, we propose the following reformulatoin of the proof in Appendix A:
>
> 1. First, we will show that we can always choose sufficiently large $\gamma=\gamma_h$ such that the tail end of the integral, $\frac{1}{2}\int_{\gamma_h}^{\infty}E_{X_\gamma |x}[ \mathcal{J}(X_\gamma)]d\gamma$,  becomes arbitrarily small for all $x$ (since it decays as $\frac{1}{\gamma_h}$; see lines 386-389, and comment i above).
>
> 2. Next we will show that, for finite $\gamma\leq \gamma_h$, we can always choose $d \equiv ||x_0- x_\gamma||^2$ so that the integrand $E_{X_\gamma |x}\left[ \mathcal{J}(X_\gamma) \right]$ is unaffected by perturbations to $p(r,x')$ in the ball $B_\epsilon(x_0)$ (lines 390-397, and comment ii above).
>
> Together, this establishes locality, since we can say that for any $\delta>0$  there exists a $d_\delta$ such that $|I(x)-\tilde I(x)|\leq \delta $ when $||x-x_0||^2\geq d_\delta$.
>
>
> iv. We will correct this typo.
>
> =============== Minor comments ==============
>
> i.  In in the main text (line 151, and Eqn 8), we claim that
> $$I(R; X)=\frac{1}{2}E_{X} \left[ I_{\text{local}}(X)\right] =\frac{1}{2}E_{X, X_\gamma}\left[ \int_{0}^{\infty} \mathcal{J}(X_\gamma)d\gamma \right].$$
> For this to follow from Eqn 5 in the main text, requires that:
> $$I(R; X)=\frac{1}{2}E_{X_\gamma}\left[ \int_{0}^{\infty}\mathcal{J}(X_\gamma) d\gamma\right]=\frac{1}{2}\int_{0}^{\infty} E_{X_\gamma} \left[ \mathcal{J}(X_\gamma) \right] d\gamma,$$
> The reviewer raises a valid question about whether this interchange of integration over $\gamma$ and $X_\gamma$ is justified.
>
> From Fubini's theorem, we can switch the order of integration so long as the integrand is Lebesgue integrable, such that $\frac{1}{2}\int_{0}^{\infty}E_{X_\gamma} \left[| \mathcal{J}(X_\gamma) |\right] d\gamma<\infty$. This is indeed the case in our setup, since $\mathcal{J}(x_\gamma)$ is always non-negative (so taking its absolute value doesn't change anything), while the integral is equal to $I(R;X)< \infty$.
>
> ii. We will replace $H$ with $h$ to denote differential entropy, as suggested.
>
> iii. From Rioul 2010, for de Brujin's identity to apply, we require that $p(X)$ and $p(X|R)$ are sufficiently smooth and decay to zero as $X$ goes to infinity, so that the Fisher information exists. We will state this in the camera ready text (see also comment 2i).
>
> =============== Additional questions & comments ==============
>
> The reviewer asked why we didn't analyse real neural data. We used a simple model of visual neurons responding to MNIST digits to show the behavior of our decomposition clearly. However, we have also applied our method to a model of retinal neurons (Klindt et al. 2017), with a diffusion model trained on natural images. This gave similar qualitative results to the MNIST dataset, and can be added to the appendix.
>
> In the new appendix, we will add a figure comparing our decomposition to  alternative decompositions (e.g. $I_{surp}$, whose feature decomposition is non-negative and non-local) to further highlight why our decomposition is both clear and interpretable.

---

> > ### Comment · Reviewer_2L2J · 2025-08-04
> >
> > I thank the authors for their detailed response, and it has helped cleared some of my misunderstandings. I have updated my score from 2 to 3.
> >
> > 1. Regarding the conditional version of the de-Brujin's indentity.
> >
> > - I appreciate the authors clearing my confusion regarding the underlying model. Indeed, my earlier concern that $p(R)$ is changing with $\gamma$ is alleviated, and I am more inclined to believe the author's result. Regardless, I still am not fully convinced that de-Brujin's identity can be applied in a straightforward manner to the conditional distribution. I personally have not been able to find a citation mentioning a conditional version of de-Brujin's identity. As I mentioned earlier, if authors could provide a citation, it would be the simplest way to alleviate my concerns.
> >
> > - Pertaining to my analysis, that was not meant to be substitute for a proof but rather some basic math to illustrate my point that application of de-Brujin's identity to conditional distribution is not trivial. Indeed, if what authors claim to be true, I believe the result for the conditional de-Brujin's identity would, by itself, be a result worth publishing, e.g., in ISIT. However, the proof of de-Brujin's identity is bit more involved, typically, requiring a solution to heat equation. In order to fully extend de-Brujin's identity to its conditional version, one would need to show a proof in a similar vein. Note that I am not saying that author's results are wrong rather, when dealing with infinite integrals and PDEs, it becomes important to carefully derive the steps, and make sure all technical details are carefully checked to ensure validity of the result. Given, the application of de-Brujin's identity to conditional distribution seems novel, it regrettably, does require additional scrutiny.
> >
> > 2. Under the revised assumptions on the pdf, many common distributions used in neuroscience would be removed, the two big ones being Poisson and Exponential distribution. Indeed, if my understanding is correct, this method would not work, if the stimulus distribution is discrete, which for example in many sensory behavior tasks (e.g., whisker stim), it might be. Technically, the assumptions also disallow the uniform distribution. I think this limits the applicability of the proposed method considerably.
> >
> > 3. I am not entirely sure where the $o(1/\gamma^2)$ is coming from. Can the authors elaborate?
> >
> > 4. Authors' reply on the proof still does not satisfactorily answer my concerns. For example, the second part of the approximation still does not fully show the proof. For example, in the second part of the proof, just showing that the error between $log(p(r|x_{\gamma}))$ and $log(\tilde{p}(r|x_{\gamma}))$ reduces for $\|x-x_0\|\gg \gamma$ is not sufficient. There is also an integral over $\gamma$. Hence, the $\int_{0}^{\gamma} E_{X_{\gamma}|x}[J(X_{\gamma})]d\gamma$ will add up the errors across the range of $\gamma$. Therefore, it needs to be shown that the integral itself is also going to zero. While, the main result might be true, in order to ascertain its veracity, it is important to be able to understand the underlying proof.

---

> ### Author Response · Authors · 2025-08-04
> **Response to each of reviewer's comments**
>
> 1. We respectfully disagree that conditioning on $R$ invalidates our use of de Bruijn’s identity. The reason is that $p(R)$ does not depend on $\gamma$, and thus we can apply de-Brujin's identity for $h(X+\sqrt{\gamma}Z|r)$ in the same way as for the `vanilla' case, by replacing $p(X)$ with $p(X|r)$, and then integrating over $p(R)$.
>
> Our result follows as a corollary of the 'vanilla' de Bruijn's identity, which states
> $$-\frac{d}{d\gamma}E_{p(X),p(Z)}[\log p(X+\sqrt{\gamma},Z)]=
> -\frac{1}{2}\mathrm{Tr}( E_{p(X)p(Z)}[\nabla^2\log p(X+\sqrt{\gamma},Z)] ).$$
> where $Z\sim\mathcal{N}(0, I)$, independent of $X$.
>
> We consider the conditional entropy, defined as, $h(X+\sqrt{\gamma}Z|R)=\int_r p(r)h(X+\sqrt{\gamma}Z|r) dr$ where
> $$h(X+\sqrt{\gamma}Z|r)=-E_{p(X|r)p(Z)}[\log p(X+\sqrt{\gamma}Z|r)].$$
> Noting (comment 1, last rebuttal) that the conditional distribution over $Z$ and $X$ can be factorised as $p(X,Z|r)=p(Z)p(X|r)$, we see that the above expression is identical to the expression for the marginal entropy, $h(X+\sqrt{\gamma}Z)$, so long as we replace $p(X)$ with $p(X|r)$ everywhere.
>
> Now, de Bruijn’s identity makes few assumptions about the form of the prior beyond smoothness and integrability. Replacing $p(X)$ with the conditional $p(X|r)$ for fixed $r$ does not violate these assumptions. Hence, for every $r$,
> $$-\frac{d}{d\gamma} E_{p(X|r),p(Z)}[\log p(X + \sqrt{\gamma},Z|r)]=-\frac{1}{2}\mathrm{Tr}(E_{p(X|r)\,p(Z)}[ \nabla^2\log p(X+\sqrt{\gamma},Z,|r)]),$$
> which follows by the same derivation as the unconditional case.
>
> Finally, integrating $h(X|r)$ over $p(R)$ gives the result for the conditional entropy derivative, $h(X|R)$ (since $p(R)$ doesn't depend on $\gamma$).
>
> 2. We assume the conditional, $p(R|X)$, is smooth with respect to $X$, so the Fisher is well-defined. This is a standard assumptions in neural coding studies, where the Fisher is widely used (Rao, 1992; Brunel \& Nadal, 1998; Yarrow et al., 2012; Ding et al., 2023). Similarly, requiring $p(X)\rightarrow 0$ at infinity mirrors influential work linking Fisher and mutual information (Wei \& Stocker, 2015; 2016; Morais et al., 2018; Kriegeskorte, 2021). While this excludes certain stimulus classes (e.g., discrete or bounded), it aligns with a broad and well-established body of neuroscience literature that models continuous, unbounded stimuli.
>
> Poisson and exponential distributions are often used in the field to describe spike counts or interspike intervals, and not the stimulus distribution. This is allowed in our work, which imposes minimal constraints on the responses.
>
> 3. Our goal is to show that at high-$\gamma$,
> $$
> \frac{1}{\gamma^2}||E[X|R,x_\gamma]-E[X|x_\gamma]||^2=\frac{1}{\gamma^2}||E[X|R]-E[X]||^2+o(1/\gamma^2).
> $$
>
> To do so, we define,
> $$ a(\gamma)=E[X|R,x_\gamma]-E[X|x_\gamma], \qquad b=E[X|R]-E[X], $$
> so that
> $$ \textrm{error} =\frac1{\gamma^2}(||a(\gamma)||^2-||b||^2)
> =\frac1{\gamma^2}(a(\gamma)-b)^T(a(\gamma)+b). $$ Then, from Cauchy–Schwarz,
> $$ |\textrm{error}| \le\frac1{\gamma^2}||a(\gamma)-b|| ||a(\gamma)+b|| \le\frac C{\gamma^2}||a(\gamma)-b||, $$
> where $C=\sup_\gamma\|a(\gamma)+b\|<\infty$ (under assumption that $p(X|x_\gamma)$ and $p(X|R,x_\gamma)$ have finite moments, for de-Bruijn's identity).  Finally, since $\|a(\gamma)-b\|\to0$ as $\gamma\to\infty$ (comment 3ia of last rebuttal), we conclude
> $$ |\textrm{error}|=\frac{C}{\gamma^2}o(1)=o\Big(\frac{1}{\gamma^2}\Big).
> $$.
>
> 4. In the paper (and last rebuttal) we show that, for $\gamma\leq \gamma_h$ we can always find sufficiently large $||x-x_0||^2$ so that $E_{X_\gamma|x}[J(X_\gamma)]$ is unaffected by perturbations of $p(r,x')$ in the region $x'\in B_\epsilon (x_0)$.
>
> The reviewer worried that, even so, the integral, $\int_0^{\gamma_h} E_{X_\gamma|x}[J(X_\gamma)]d\gamma$ could be affected by the perturbation, since the errors (integrated over $\gamma$) could 'add up'. The short response is that, since we integrate in a finite range of $\gamma$, we can always choose  $\gamma$ where the difference (between Fisher with respect to perturbed/unperturbed distribution) is largest, and then use this to upper bound the integral. We show this below.
>
> We define  $c(x,\gamma)=E_{X_\gamma|x}[\mathcal{J}(X_\gamma)]-E_{X_\gamma|x}[\mathcal{\tilde{J}}(X_\gamma)]$, with $\tilde{\mathcal{J}}$ obtained using the perturbed distribution. We need to show that $\Big|\int_{0}^{\gamma_h}E_{X_\gamma|x}[\mathcal{J}(X_\gamma)]d\gamma-\int_{0}^{\gamma_h}E_{X_\gamma|x}[\mathcal{\tilde{J}}(X_\gamma)]d\gamma \Big|=|\int_{0}^{\gamma_h} c(\gamma,x)d\gamma|$
> can be made arbitrarily small for large $||x-x_0||^2$.
>
> We can bound this integral from above:
> $$\Bigg|\int_{0}^{\gamma_h}c(\gamma,x)d\gamma \Bigg| \leq \int_{0}^{\gamma_h} |c(\gamma,x)|d\gamma \leq \gamma_h \sup_{\gamma\in { \{0,\gamma_h \}}} |c(x, \gamma)|.$$
> Thus, if we can choose $||x-x_0||^2$ to ensure that $c(x,\gamma)$ is arbitrarily small for all $\gamma\leq\gamma_h$, then the upper bound on the integral goes to zero, as required.

---

> ### Comment · Reviewer_2L2J · 2025-08-04
>
> Thanks, the final theoretical arguments provided by the authors seem correct. Indeed, it is this level of theoretical rigor that is needed in order to provide a sound theoretical paper, which was sorely lacking in the first draft. I would advise the authors to maintain this level of theoretical rigor when writing a theoretical work. I have increased the score to 4, although I still think a much more thorough comparison with other method should be included in the work, with the mention that many of the other methods do work with discrete and bounded random variables, whereas this method does not. Note that in many important neuroscientific questions, the stimuli can be Poisson, as we want to understand the influence of one brain area on another, not just external stimuli

---

### Official Review · Reviewer_f3nf · 2025-06-23

**Clarity:** 4
**Significance:** 3
**Originality:** 3
**Rating:** 5
**Confidence:** 3

**Summary:**

The paper derives a stimulus-dependent decomposition of mutual information, $I_{local}$. For this the mutual information between a noise corrupted stimulus and neural population response is expressed via an expectation over the Fisher information given the stimulus distribution. To obtain a stimulus-specific estimate of the information the authors propose to condition the Fisher information on a fixed stimulus, rather than averaging over the full stimulus distribution. The authors show that this decomposition respects the axioms of completeness, additivity, positivity and locality.

Since the Fisher information can be written in terms of a mean squared-score, it can be approximated by training denoising diffusion models to sample from the prior and posterior distributions. This also means that $I_{local}$ can be estimated for high dimensional stimuli.

The authors compare to other existing stimulus-specific decompositions of mutual information and demonstrate their method allows them to estimate pixel-wise information on MNIST for a population of visual neurons.

**Questions:**

- In L144, where does the mean-squared score come from? I am not familiar with this formulation. If it is obvious would you mind giving a brief explanation? Otherwise it would be nice to have a citation there.
- Does this method also work for discrete responses?
- Are there other features / axioms of information decompositions like uniqueness that the proposed method violates?
- L542. What is the purpose of A and why did the method require an A=2? If you could give a reason for this choice, that would be helpful.

**Ethical Concerns:**

["NO or VERY MINOR ethics concerns only"]

**Final Justification:**

The paper is well written, technically solid and makes a notable contribution that I would like to see presented at the conference. The authors also have answered all my questions during the rebuttal and clarified any remaining concerns I had. I recommend this paper for acceptance.

**Limitations:**

Some limitations where discussed, although potentially a bit brief. The authors could expand on this, but I lack the knowledge to propose additional limiting factors to discuss.

**Paper Formatting Concerns:**

## Typos
- L385 "a matrix"
- L395 "= ("
- L427 Inconsistent ref / labeling Supp Fig 1. vs. Fig 5.

**Quality:**

3

**Strengths And Weaknesses:**

## Strengths
- The paper is well structured and clearly written. Despite not being an expert on decomposition of mutual information, I was able to follow the derivation and arguments quite well.
- The derived decomposition seems very useful in practice, especially because it can scale to high dimensional stimulus paradigms such as those commonly found in visual decoding experiments.

## Weaknesses
- The choice of colors in Figure 2 are a bit confusing. I feel like in panel A you could have added a Gaussian (black) and colored the bimodal in red (maybe even all of them as solid lines). Also the red in panel A could suggest it has something to do with the bimodal distribution. You might want to consider changing it to something else.
- The proposed decomposition $I_{local}$ seems most similar to that of Kong et al. 2024, yet it not compared to in the main paper (Sec. 3 for example). Why is this the case? Which are the axioms it violates (L408)? How do they compare (L405, visually, similar to Fig 1.)? Related: in L425 instead of explaining, it would be nice to be able to see this in a plot.
- While the behavior of the Fisher information for a Gaussian was mentioned in L197 it would be nice to also show this as part of Fig. 1B
- L218 a reference to the data processing inequality would be nice (Remark 1). Also which are the concrete methods that do not respect this inequality?
- L239 If I understand this correctly, this is just a LNP model. Could you state that in the text?

---

> ### Author Rebuttal · Authors · 2025-07-28
>
> We thank reviewer 1ojo for the time taken to write a detailed review. Below we go through the specific points raised, and questions.
>
> ======= Weaknesses =======
>
> 1. We decided to only show the bimodal distribution in the top part of Fig 2A, since the middle and lower plots only show the posteriors obtained with this prior distribution. We feel that adding the gaussian distribution on top of these plots as well would be confusing. However, if the reviewer thinks it will improve clarity,  we could additional panel beforehand, similar to the current panel 2A but showing the behaviour with a gaussian prior.
>
> We agree that it would be a good idea to change the colour of the red plots in panel 2A, so as to not confuse with panels 2B-D where red colour indicates the bimodal distribution. We can do this for the camera ready submission.
>
> 2. Kong et al. proposed two different decompositions of the mutual information $I(R;X)$, into functions of both the response and stimulus, $I(r,x)$ (Eqns 18 & 19, Appendix B). In contrast, we sought to decompose the information into a function of just the stimulus, $I(x)$. We thus decided to compare our decomposition in the main paper to previously proposed stimulus-dependent decompositions of the mutual information that take the same form as ours, and only discuss Kong et al.'s method textually, while keeping more detailed comparison with their work to appendix B.
>
> In appendix B, we looked at the behaviour of stimulus dependent decompositions, obtained by averaging Kong et al's decompositions over $p(R|x)$. We show that when we do this with first of their decompositions (Eqn 20) it becomes equivalent to $I_{surp}(x)$ (Eqn 3), investigated in the main text (e.g.~Fig 1E), which violates the locality axiom (Table 1). The second  decomposition (Eqn 21), on the other hand, violates the additivity axiom, as we show in the Appendix B, Figure 5. This is discussed in lines 435-440 of Appendix B and lines 123-127 and 262-271 of the main text.
>
> 3. This is a good suggestion. We can add the Fisher information to Figure 1 for the camera-ready version.
>
> 4. The data processing inequality is a standard result and can be found for example in most textbooks on information theory (e.g. Cover; Thomas (2012). Elements of information theory). We will add this reference to the camera ready paper. In figure 3 of our paper we show that, while the data processing inequality is always respected on average for the mutual information, both the stimulus specific information (Eqn 2) and specific surprise (Eqn 3) violate the data processing inequality point-wise,  for individual stimuli (3D-E).
>
> 5. Yes, we use is an LNP model. This was written in the figure caption (Fig 4), but we can also add it to the main text for clarity, for the camera ready version.
>
> ========= Questions =========
>
> 1. The mean-square expression (Eqn 6) arises from the standard identity for Fisher Information:
> $$\mathcal{J}(x) = -E_{R|x} \left[ \nabla_x^2 \log p(R|x) \right] = \mathbb{E}_{R|x} \left[ \left( \nabla_x \log p(R|x) \right)^2 \right].$$
> This is a standard result, covered in texts such as "Lehmann and Casella (1998), Theory of Point Estimation". We will add this citation in the camera-ready version.
>
> 2. Yes, our method works for both discrete and continuous responses. For example, our 1-d simulations (Figs 1-3) assume that $R$ is continuous, with gaussian noise, while our high-d simulation (Fig 4) assumes discrete responses, with Poisson noise.
>
> 3. Unlike the coordinate-invariant stimulus-specific information (CiSSI; Eqn 4), our decomposition is not invariant under arbitrary invertible non-linear transformations of the stimulus variable $X$. Achieving such invariance would require sacrificing our locality axiom—an axiom we consider more important for interpretability, for the reasons discussed in the paper. We are happy to clarify this in the paper if the reviewer feels it would improve transparency.
>
> 4. The reviewer drew our attention to a mistake in the appendix: the constant $A$ should appear as a multiplicative factor in Eqn 24 (not Eqn 23), and its value should be 40, not 2. This constant reflects the maximum spike count in our model, chosen to match the order of magnitude observed in V1 recordings over a 0.75 s window, where neurons typically reach firing rates of ~50 Hz. Importantly, our qualitative results are robust to this choice: they remain unchanged if $A$ is rescaled by a factor of 2 for example.

---

> > ### Comment · Reviewer_f3nf · 2025-08-07
> > **response to author's rebuttal**
> >
> > I thank the authors for their detailed reply to my questions, their clarifications and the pointers to relevant references. This has helped me better understand some parts of the manuscript. The authors have clarified and addressed all of my concerns.
> >
> > Comments:
> > - I will leave potential changes to figure 2 up to the authors. I agree with the authors there are pros and cons to showing the Gaussian prior or not.
> > - It would be great if the authors could add a sentence about their decomposition not being invariant under arbitrary invertible non-linear transformations for the camera ready version. While one would need to sacrifice locality for this, I still feel that this is a useful piece of information for the reader.
> > - I believe adding a similar comment about how you chose $A$ would be a neat piece of information for potential users to have.
> >
> > The paper makes a notable contribution that I would be thrilled to see presented at the conference. I will recommend this paper for acceptance and therefore maintain my score.

---

### Official Review · Reviewer_U4Li · 2025-07-01

**Clarity:** 2
**Significance:** 2
**Originality:** 2
**Rating:** 4
**Confidence:** 4

**Summary:**

This paper presents a method for decomposing mutual information between stimuli and neural responses into stimulus- and feature-specific components. By introducing three core axioms - additivity, positivity, and locality - the paper derives a meaningful decomposition. Empirical results demonstrate the approach's effectiveness in some simple scenarios.

**Questions:**

1. The completeness axiom relies on a diagonal approximation of the Fisher information matrix. How would alternative approximations—such as block-diagonal or Kronecker-factored variants—affect the satisfaction of the axioms (completeness, locality, positivity, additivity)?

2. What are the key challenges in applying this decomposition framework to high-dimensional or natural image datasets? Are the axioms still meaningful or applicable in such contexts?

3. How does the computational cost of the decomposition scale with the number of neurons or layers in modern neural networks? Is the method tractable for large-scale models?

4. The paper draws connections between Fisher information and diffusion processes in theory (Section 4). In the MNIST experiments, can the authors clarify the underlying relationship between theory and empirical observations ？

**Ethical Concerns:**

["NO or VERY MINOR ethics concerns only"]

**Final Justification:**

All my concerns have been addressed, including: 1) overly restrictive locality assumption; 2). insufficient empirical validation

**Limitations:**

see weakness and questions

**Quality:**

2

**Strengths And Weaknesses:**

***Strengths*** :

The paper introduces an interesting axiomatic framework for decomposing neural information, offering a useful perspective that could advance probing and interpretability techniques in neural networks.

***Weaknesses*** :

1. The assumption of locality appears too restrictive for practical applications. It implicitly requires neural representations to be disentangled, which rarely holds in practice - neither in high - performing artificial neural networks nor in biological brains. For instance, conjunctive neurons, which encode multiple features jointly, are prevalent and critical for generalization in biological brains. This undermines the generality and applicability of the proposed axioms.

2. While the paper claims to quantify stimulus or stimulus feature-wise contributions to encoded information, the experimental validation is limited to toy settings. There is no comparison with established interpretability methods on more realistic datasets. For instance, the MNIST experiment lacks ablation studies and quantitative evaluation, making it difficult to assess the practical value and necessity of the three proposed axioms.

---

> ### Author Rebuttal · Authors · 2025-07-29
>
> We thank reviewer 1ojo for the time taken to write a detailed review. Below we go through the specific points raised, and questions.
>
> ============ Weaknesses ==========
>
> 1. We believe there is some confusion regarding the locality principle, which we will try to clarify here.
>
> Our locality assumption imposes a constraint on  the information decomposition, not the neural network under study. Specifically, it does *not* impose any constraints on how neurons respond to stimuli—e.g., it does not assume disentangled or independent responses.
>
> The locality axiom states that the stimulus-dependent information, $I(x)$, should depend only on how neurons respond to stimuli near $x$ in stimulus space—not on their responses to distant stimuli $x'$. This is crucial for interpretability. For example, consider two neural populations: one responding to trees, the other to faces. Without locality, the information attributed to a face stimulus, $I(x_{\text{face}})$, could incorrectly depend on how tree-selective neurons respond, even if the two populations are functionally independent.
>
> Conversely, if the populations are entangled—e.g., responses to faces are modulated by responses to trees—this coupling will naturally be reflected in $I(x)$, as it should.
>
> In summary, the locality axiom is not restrictive, but rather ensures that $I(x)$ provides a meaningful, interpretable decomposition of information encoded by the network. We can add an additional sentence to the manuscript to clarify this if the reviewer believes it will be helpful.
>
> 2i: comparison with litterature
>
> We respectfully clarify that we are not aware of established interpretability methods that directly address our problem. Classic saliency methods like gradient attribution, integrated gradients, or CAM have been designed for discriminative models (classifiers or regressors, with examples in computational neuroscience as well see Tanaka et al, NeurIPS 2019, and Maheswaranathan et al, Neuron 2023) and single-output settings. They cannot be straightforwardly applied to mutual information or generative sensory coding tasks (see, for example, Zheng et al. ICLR 2024, for a discussion on the challenges of developing interpretability methods for generative diffusion models).
>
> For our problem, of decomposing the mutual information into contributions from each stimulus/feature, the only direct baselines are previously proposed stimulus-specific information measures such as $I_{\text{SSI}}$, the specific information $I_{\text{sp}}$, surprise $I_{\text{surp}}$, and coordinate-invariant SSI $I_{\text{CiSSI}}$, which we extensively discuss and theoretically compare in Section 3 and Table 1. However, these previous measures are either intractable in high-dimensions or violate key interpretability properties (e.g. $I_{\text{sp}}$ and $I_{\text{SSI}}$ can become negative and all these measures are non-local, meaning a change in model behaviour at one stimulus can affect attributions everywhere; see table 1, Fig. 1C–F and Fig. 3D–E). We illustrate this in simple situations, showing that our measure reduces to known quantities in special cases (e.g. it recovers the Fisher information in the small-noise limit and matches total mutual information when averaged over stimuli) and by contrasting its qualitative behaviour against previous measures in simplified scenarios (Figures 1–3).
>
> To address the reviewers comment, asking for a comparison with previous methods on high-d dataset, we will add a new Appendix, showing how a feature-wise decomposition of $I_{surp}(x)$ (which we can estimate using a diffusion model, as shown in Appendix B) behaves on the MNIST dataset. Since the feature-wise decomposition of $I_{surp}(x)$ can be both negative and non-local, this should allow us to further highlight the importance of our axioms for interpretability on high-d datasets.
>
> 2ii: application to realistic data-sets
>
> To address the reviewers' comments, we have performed additional simulations using a model fitted to real retinal data (Klindt et al., NeurIPS 2017), and a diffusion model trained on natural images. The qualitative behaviour in this setting closely mirrors that observed with MNIST: the decomposition highlights regions of the image with high local spatial contrast. This can be included in an Appendix for the camera ready version.
>
> ============ Questions ============
>
> 1. To clarify, we do not make any assumptions about the Fisher Information matrix (e.g. we don't assume that it is diagonal). Instead, the multivariate de Brujin identity relates the derivative of the entropy (with respect to the variance of an additive gaussian noise perturbation) to the trace of the Fisher information, as shown in Eqns 5. We will add a citation (to Rioul et al. 2010) to clarify this.
>
> 2. Figure~4 illustrates the behavior of the decomposition in a moderately high-dimensional setting (images with $32 \times 32 = 1024$ pixels and 50 neurons). In the revised appendix, we will include additional results from simulations using a model trained on real retinal data, along with a diffusion model trained on natural images. For natural images, we used 64x64 resolution, compared to 32x32 for MNIST, while the training dataset size remained consistent at 60,000 samples, equivalent to MNIST.  In our experiments, we needed neither more image samples nor additional diffusion steps compared to MNIST to identify informative areas in natural images.
>
> In high-d settings, our locality axiom becomes even more critical than in the 1D examples shown in Figs. 1-3. Without locality, the attribution $I(x)$ could be influenced by the network's response to *any* stimulus—even those far from $x$ in stimulus space. In low dimensions, one can visualize how $I(x)$ varies across stimuli, making such non-local effects apparent. But in high dimensions, this is no longer feasible: one cannot easily determine which stimulus-response pairs contribute to the attribution at a given $x$. Enforcing locality ensures that $I(x)$ reflects only the behaviour of the system in a neighborhood around $x$, preserving interpretability. To show this we will include an additional figure panel in the Appendix, showing the behaviour of $I_{surp}(x)$ (which doesn't respect locality) on a high-d dataset (see previous comment).
>
> 3. The computational cost of our decomposition method depends on the cost of training a conditional diffusion model, that is able to sample from the posterior distribution $p(X|r)$. In our case, we used a multilayer perceptron that transforms the model neural responses into a 256-embedding, which we feed as an input to the conditional diffusion model. The complexity thus scales linearly as $O(n_{neurons}\times 256)$. In practice, since conditional diffusion models can be trained on high-d inputs, the bottleneck for analysing real biological neural populations will typically come from the experimental dataset (i.e. how many neurons were recorded), rather than the diffusion model.
>
> 4. In section 4, the relation between the Fisher information about a noise corrupted stimulus, $\mathcal{J}(x_\gamma)$, and the expectations, $\hat{x}(X|x_\gamma,r)$ & $\hat{x}(X|x_\gamma)$, follows directly from Tweedie's law, which relates the score in a diffusion model to these conditional expectations. Tweedie’s law is a standard result which has been used extensively  in previous work on high-d datasets (e.g. Kadkhodaie & Simoncellie Neurips 2021).
>
> In Figure 4 we plot the Fisher information (with respect to the true stimulus, not the noise corrupted stimulus), $\mathcal{J}(x)$, alongside $I_{local}(x)$, for the  MNIST digits. In this setting, where the stimulus dimension is higher than the number of neurons, the Fisher information has limited relevance for measuring how well we can extract information about the stimulus from neural responses (e.g. the Cramer-Rao bound, which relates the Fisher to the optimum decoding error, diverges). In contrast, due to the completeness axiom, $I_{local}(x)$ is directly related to the total information encoded by the network, $I(R;X)$.

---

> > ### Comment · Reviewer_U4Li · 2025-08-04
> >
> > Thank you for the detailed response. My concerns have been addressed, and I have increased the overall score from 3 to 4.

---

### Official Review · Reviewer_1oJo · 2025-07-02

**Clarity:** 3
**Significance:** 4
**Originality:** 4
**Rating:** 6
**Confidence:** 3

**Summary:**

The paper proposes a new measure of neuronal sensitivity generalizing the Fisher information, focusing on the separate contributions of specific stimulus features or dimensions to evoked neuronal activity. After giving the criteria that such a measure should meet and defining a specific measure that satisfies those desiderata, the paper shows how to use both unconditional and conditional denoising diffusion models to actually compute the defined measure.  An initial experiment shows that the measure appears to serve its purpose fairly well when computed on MNIST.

**Questions:**

In lines 59-60, the statement of Axiom 1, what stimulus distribution does the expectation integrate over?

Since the problem of estimating the local information is ultimately one of approximating an integral over noise levels, do the authors know something other than a diffusion model that could potentially compute the approximation?  If a better neural architecture than diffusion models were found for denoising, could that be integrated into the estimation here?

What would happen if I_{\mathrm{local}}(\mathbf{x}) were evaluated over spontaneous or intrinsically generated neuronal spiking, rather than over stimulus-evoked activity?

**Ethical Concerns:**

["NO or VERY MINOR ethics concerns only"]

**Final Justification:**

The authors have specifically justified their motivation in using conditional diffusion models and added an experiment on natural data rather than MNIST.  I would like to champion this paper.

**Limitations:**

yes

**Quality:**

4

**Strengths And Weaknesses:**

Strength: axioms are well-motivated for extending the intuitive properties of Fisher information and mutual information

Strength: a method is given to approximately compute the information-theoretic quantity defined using extent computational tools

Weakness: experiments are conducted entirely on rate-coded neurons rather than spiking ones.
Weakness: related to the above, experiments are on model neurons rather than on a real dataset

---

> ### Author Rebuttal · Authors · 2025-07-28
>
> We thank reviewer 1ojo for the time taken to write a detailed review. Below we go through the specific points raised, and questions.
>
> === Highlighted weaknesses===
>
> - We chose rate coding neurons in our simulations for simplicity, to better isolate and illustrate the core properties of the decomposition. However, the method is general: the mathematical framework applies equally to spiking neurons. Just as Shannon information theory is agnostic to the neural code, our decomposition can be applied to either rate- or spike-based representations.
>
> - We presented results from a simple LNP model responding to MNIST digits. This setup was sufficient to demonstrate the qualitative properties of the decomposition and its scalability to high-dimensional inputs and multi-neuron populations.
>
> To address the reviewer’s concern about the inclusion of real-world data, we have tested our method on a model trained with real retinal data (Klindt et al., NeurIPS 2017), in addition to the simpler MNIST dataset. We chose to highlight the MNIST dataset in the main text to demonstrate our method’s applicability to a broad audience using a well-known, accessible dataset. However, we can now include results from the retinal data model in a new Appendix for the revised manuscript, to demonstrate the robustness of our approach across diverse datasets, directly addressing the reviewer’s concern
>
> === Questions ====
>
> 1. As with Shannon mutual information, our decomposition depends on the assumed stimulus distribution, $p(X)$. In practice, when modeling neural coding under naturalistic conditions, it is standard to use the natural image distribution (e.g., Simoncelli & Olshausen, 2001). That said, our framework does not require this assumption; any stimulus distribution can be used depending on the experimental context.
>
> 2. Diffusion models are a natural fit for our decomposition because they directly provide the required conditional expectations, $E[X \mid X_\gamma]$ and $E[X \mid X_\gamma, R]$, across a range of noise levels $\gamma$. However, other architectures capable of similar denoising (e.g., Kadkhodaie & Simoncelli's "bias-free" network, NeurIPS 2019) could also be used. Our decomposition is not tied to a specific model class.
>
> 3. Our work focuses on decomposing the mutual information between a stimulus $X$ and a response $R$, i.e., $I(R; X)$, which is not well-defined in the absence of a stimulus. For spontaneous or intrinsic activity, where $X$ is undefined or not present, the current decomposition does not apply.

---

### Decision · Program_Chairs · 2025-09-17

**Decision:**

Accept (spotlight)

**Comment:**

a) The authors propose an axiomatic construction of stimulus specific information and propose a way to estimate it using diffusion models.

b) The axioms are sound and justified. The authors details other previously proposed definitions of stimulus specific information and show that they do not satisfy all axioms. They propose a method to construct a version that satisfies all axioms and use the method to estimate the information capture by a model of visual neurons (using MNIST).

c) No neural data. Initial concerns about the mathematical rigor (resolved!)

d) Estimating mutual information and other type of information theoretic quantity is a difficult task. The authors provide a axiomatic definition and a way to estimate the quantity that verifies their axiom. This paper deserves to be discussed as the proposed axioms can be questioned. A spotlight is great opportunity to trigger some discussion at the conference.

e) Two reviewers were initially skeptical about acceptance. The rebuttal period allowed to answer to their main concerns and both reviewers change their mind and are now in favor of accepting the paper.